# Lpcat3-dependent production of arachidonoyl phospholipids is a key determinant of triglyceride secretion

Xin Rong[1], Bo Wang[1], Merlow M Dunham[2,3], Per Niklas Hedde[4,5], Jinny S Wong[6], Enrico Gratton[4,5], Stephen G Young[7], David A Ford[2,3], Peter Tontonoz[1]*

[1]Department of Pathology and Laboratory Medicine, Howard Hughes Medical Institute, University of California, Los Angeles, Los Angeles, United States; [2]Department of Biochemistry and Molecular Biology, Saint Louis University, St. Louis, United States; [3]Center for Cardiovascular Research, Saint Louis University, St. Louis, United States; [4]Laboratory of Fluorescence Dynamics, Biomedical Engineering Department, University of California, Irvine, Irvine, United States; [5]Center for Complex Biological Systems, University of California, Irvine, Irvine, United States; [6]Electron Microscopy Core, Gladstone Institute of Cardiovascular Disease, San Francisco, United States; [7]Division of Cardiology, Department of Medicine, University of California, Los Angeles, Los Angeles, United States

**Abstract** The role of specific phospholipids (PLs) in lipid transport has been difficult to assess due to an inability to selectively manipulate membrane composition in vivo. Here we show that the phospholipid remodeling enzyme lysophosphatidylcholine acyltransferase 3 (Lpcat3) is a critical determinant of triglyceride (TG) secretion due to its unique ability to catalyze the incorporation of arachidonate into membranes. Mice lacking *Lpcat3* in the intestine fail to thrive during weaning and exhibit enterocyte lipid accumulation and reduced plasma TGs. Mice lacking *Lpcat3* in the liver show reduced plasma TGs, hepatosteatosis, and secrete lipid-poor very low-density lipoprotein (VLDL) lacking arachidonoyl PLs. Mechanistic studies indicate that Lpcat3 activity impacts membrane lipid mobility in living cells, suggesting a biophysical basis for the requirement of arachidonoyl PLs in lipidating lipoprotein particles. These data identify Lpcat3 as a key factor in lipoprotein production and illustrate how manipulation of membrane composition can be used as a regulatory mechanism to control metabolic pathways.

*For correspondence: ptontonoz@mednet.ucla.edu

## Introduction

Phospholipids (PLs) are important components of biological membranes and also serve as precursors for the generation of diverse signaling molecules (*Spector and Yorek, 1985*). In mammalian cells PLs synthesized de novo undergo further remodeling through deacylation by phospholipases and the subsequent and reacylation by lysophospholipid acyltransferases (Lpcats). Membrane PLs ultimately reach an equilibrium in which the majority of PL species contain a saturated acyl chain at the *sn*-1 position and an unsaturated chain at the *sn*-2 position. The Lpcat-dependent remodeling process is essential for the diversity and asymmetric distribution of acyl chains because the de novo PL synthesis pathway has little substrate specificity (*Yamashita et al., 2014*).

We previously identified the sterol-activated nuclear receptor LXR as an integrator of cellular lipid levels and membrane PL composition. LXR controls the expression of Lpcat3, which is the most abundant Lpcat family member in liver and intestine. Cell-based assays suggest that Lpcat3 preferentially catalyzes the synthesis of phosphatidylcholine (PC) species containing an unsaturated

**eLife digest** Living cells are surrounded by a membrane that forms a barrier between the cell and its external environment. This membrane is largely made up of a variety of molecules known as lipids. The particular lipid molecules found in a cell membrane strongly influence its mobility, flexibility and other physical properties.

The liver and intestine can package lipids gained from the diet into molecules called lipoproteins, which are released into the bloodstream for use by the body. An enzyme known as Lpcat3 is found in high levels in the liver and intestine and it appears to be involved in the production of lipoproteins. Altering the amount of Lpcat3 in cells can change the types of lipids found in the cell membranes, connected to the production of lipoproteins.

Rong et al. studied newborn mice that were missing the Lpcat3 protein in either the liver or intestine. Mice lacking Lpcat3 in the intestine had higher levels of lipids inside their intestine cells and grew more slowly than normal mice. Mice lacking Lpcat3 in the liver also accumulated lipids in their cells and their bloodstream had lower levels of lipids that contain a molecule called arachidonic acid than normal mice. Further experiments showed that the loss of Lpcat3 reduces the ability of lipids to move within the cell membrane.

The experiments show that Lpcat3 plays a key role in attaching arachidonic acid to membrane lipids to promote the release of lipoproteins into the bloodstream. Rong et al.'s findings reveal that changing the type of lipids in the cell membrane plays an important role in regulating the levels of lipids in the blood.

fatty acyl chain at the sn-2 position (*Hishikawa et al., 2008*; *Zhao et al., 2008*), but the importance of Lpcat3 activity for membrane PL composition in vivo remains to be established. We found that acute overexpression or knockdown of Lpcat3 in cultured cells or mouse liver altered the distribution of PL species, particularly those containing unsaturated fatty acyl chains (*Rong et al., 2013*). Moreover, we showed that the ability of the LXR-Lpcat3 pathway to promote unsaturation of membrane lipids was protective against ER stress and inflammation in the setting of cellular lipid excess. However, the in vivo relevance of LXR-dependent modulation of endogenous PL composition for systemic lipid homeostasis was unclear.

Moreover, the larger topic of the regulatory potential of dynamic phospholipid remodeling has been largely unexplored. It has been reported that modifications of PL composition influence a range of cellular processes (*Holzer et al., 2011*; *Pinot et al., 2014*). However, most analyses of the consequences of altered PL fatty acyl composition have been performed in purified membrane systems or have involved treating cells with high levels of exogenous lipids. Such experimental manipulations are unlikely to accurately model physiologic perturbations in membrane composition. The recognition that Lpcat3 activity can be regulated by cellular lipid status through LXRs raises the possibility that Lpcat3 activity could contribute to some of the well-documented effects of LXR on systemic lipid homeostasis.

The rate of lipoprotein production has been linked with the availability of PC, but the mechanisms underlying this link are not clear (*Vance, 2008*; *Abumrad and Davidson, 2012*). PC is the major PL component of lipoproteins (*Ågren et al., 2005*). It has been reported that active PC de novo synthesis is necessary for the biogenesis and/or secretion of very low-density lipoprotein (VLDL) from hepatocytes (*Vance, 2008*). However, impaired de novo PC synthesis impacts all the PC species and reduces the total PC content of cellular membranes. Whether a specific PC species is selectively required for hepatocyte lipoprotein production is unclear. In the intestine, it has been proposed that PC may enhance lipid uptake by enterocytes and/or promote chylomicron assembly and secretion (*Tso et al., 1977*, *1978*; *Voshol et al., 2000*). Luminal PC is not absorbed intact, but is hydrolyzed into lysophosphatidylcholine (Lyso-PC) in the lumen and subsequently re-acylated within enterocytes by Lpcat enzymes (*Nilsson, 1968*; *Parthasarathy et al., 1974*). The mechanistic role of specific PL species in lipid transport and lipoprotein production has been difficult to address due to an inability to selectively manipulate membrane PL composition in vivo. Interestingly, the LXR pathway has been reported to promote hepatic triglyceride (TG) secretion by the liver (*Okazaki et al., 2010*). The possibility that PL remodeling contributes to this effect has not been tested.

We demonstrate here that Lpcat3 is uniquely required for the incorporation of arachidonic acid into membranes in vivo, and that an absence of arachidonoyl PLs profoundly affects lipid transport and lipoprotein production. Biophysical, electron microscopy (EM) and biochemical studies indicate that Lpcat3-dependent production of arachidonoyl PLs is important for lipid movement within membranes and for the efficient lipidation of apoB–containing lipoproteins. We also show that induction of Lpcat3 activity is required for the ability of LXRs to promote hepatic VLDL production. These data identify Lpcat3-dependent phospholipid remodeling as a critical, LXR-regulated step in TG secretion, and suggest that this step might be further explored as a strategy to treat hyperlipidemias.

## Results

To examine the consequence of Lpcat3 deficiency in vivo, we generated Lpcat3-deficient mice from targeted ES cells (*Figure 1A*). The targeted allele was 'conditional-ready', making it possible to create both global and tissue-specific knockout mice. The global knockout mice (i.e., homozygous for the targeted allele) showed markedly reduced levels of *Lpcat3* transcripts in liver and intestine (*Figure 1B,C*).

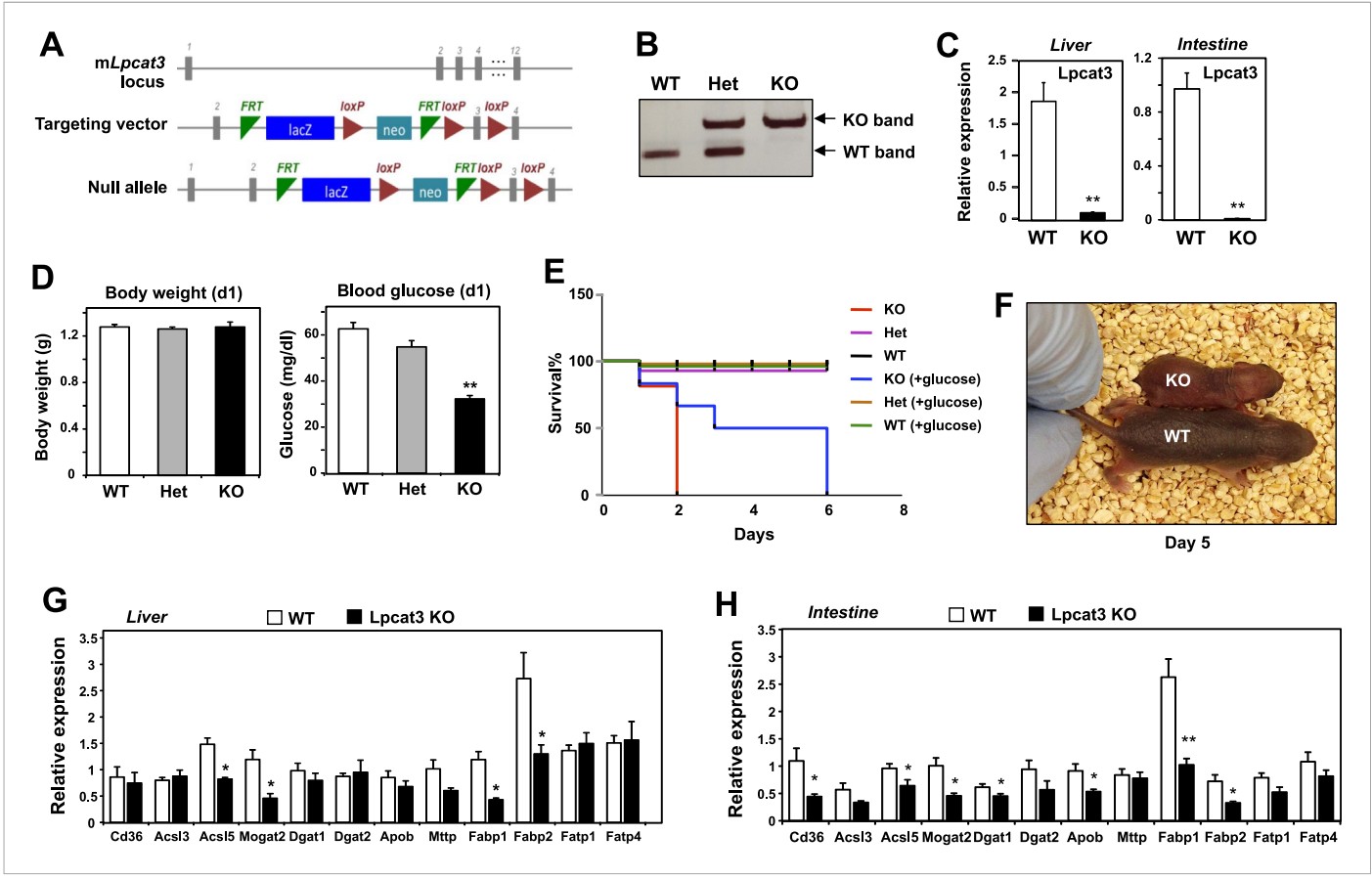

**Figure 1**. Generation and analysis of global *Lpcat3* knockout mice. (**A**) Strategy for generating global *Lpcat3* knockout mice. A 'knock-out first/conditional-ready' gene-targeting vector was used to generate targeted cells. A gene-trap cassette is located between the two FRT sites. *LacZ*, β-galactosidase; *neo*, neomycin phosphotransferase II. (**B**) Genotyping of *Lpcat3*$^{+/+}$ (WT), *Lpcat3*$^{+/-}$ (Het), and *Lpcat3*$^{-/-}$ (KO) mice. Genomic DNA was prepared from tail biopsies, and PCR products were separated on a 1% agarose gel. (**C**) Expression of Lpcat3 in liver and small intestine of newborn *Lpcat3*$^{-/-}$ and *Lpcat3*$^{+/+}$ pups. Gene expression was quantified by real-time PCR ($n \geq 5$/group). Values are means ± SEM. (**D**) The body weight and blood glucose of Lpcat3$^{+/+}$ (WT), Lpcat3$^{-/+}$ (Het), and Lpcat3$^{-/-}$ (KO) newborn pups (n ≥ 8/group). Values are means ± SEM. (**E**) Kaplan–Meier survival curve of *Lpcat3*$^{+/+}$ (WT), *Lpcat3*$^{-/+}$ (Het) and *Lpcat3*$^{-/-}$ (KO) pups after birth ($n \geq 20$ mice/group). The neonatal lethality can be delayed by injection of 50 µl 10% glucose solution once per day after born. 5 *Lpcat3*$^{+/+}$ (WT), 9 *Lpcat3*$^{-/+}$ (Het) and 6 *Lpcat3*$^{-/-}$ (KO) mice were used in the rescue experiment. (**F**) Representative photograph of *Lpcat3*$^{+/+}$ (WT) and *Lpcat3*$^{-/-}$ (KO) pups after 5 days of glucose injections. (**G–H**) Gene expression in livers (**G**) and small intestines (**H**) of *Lpcat3*$^{+/+}$ (WT) and *Lpcat3*$^{-/-}$ (Lpcat3 KO) newborn pups. Gene expression was quantified by real-time PCR (n ≥ 5/group). Values are means ± SEM. Statistical analysis was performed using Student's *t*-test (**C**, **G** and **H**) and one-way ANOVA with Bonferroni post-hoc tests (**D**). *p < 0.05; **p < 0.01.

Global *Lpcat3*<sup>−/−</sup> mice on a C57BL/6 background were born at the expected Mendelian frequency, and their weights were indistinguishable from WT mice at birth (*Figure 1D*, *Table 1*). However, the blood glucose levels of *Lpcat3*<sup>−/−</sup> mice were very low at birth, and none survived beyond day 2 (*Figure 1D,E*). *Lpcat3*<sup>−/−</sup> pups survived for up to 6 days when given subcutaneous glucose injections, but the pups did not thrive and invariably died (*Figure 1E,F*). Analysis of gene expression in liver and small intestines of the pups revealed changes in a number of genes linked to lipid metabolism, some of which were common in both tissues (*Figure 1G,H*). Although these changes appeared consistent with a role for Lpcat3 in lipid metabolism, it was impossible to exclude the possibility that these gene-expression alterations were simply due to the extremely poor health of the mice. We therefore turned our attention to tissue-selective knockout mice, with the hope that we could obtain viable mice and decipher the function of Lpcat3.

We generated a conditional knockout allele (*Lpcat3*<sup>fl</sup>) by breeding the global heterozygous knockout mice with mice expressing FLPe recombinase (*Rodriguez et al., 2000*). Mice harboring the 'floxed' *Lpcat3* allele were then crossed with albumin-Cre transgenic mice to create liver-specific Lpcat3 knockout mice (here designated 'L-Lpcat3 KO'; *Figure 2A*). In contrast to the global *Lpcat3* knockout mice, L-Lpcat3 mice were born at the expected Mendelian frequency, survived to adulthood, and appeared (at least by external inspection) to be indistinguishable from control (homozygous floxed, Cre-negative) mice (*Table 2* and data not shown). Expression of *Lpcat3* transcripts in whole liver from L-Lpcat3 KO mice was markedly reduced (*Figure 2B*). The residual expression of *Lpcat* mRNA in the liver of Lpcat3 KO mice was likely due to persistent expression of Lpcat3 in cell types that do not express the albumin-Cre transgene (Kupffer cells, endothelial cells). Consistent with that idea, *Lpcat3* expression was reduced by >90% in primary hepatocytes from L-Lpcat3 KO mice (*Figure 2B*). Unfortunately, we were unable to measure levels of Lpcat3 protein because specific antibodies are not currently available. We observed no compensatory upregulation of *Lpcat1* or *Lpcat2* in livers of L-Lpcat3 KO mice (*Figure 2B*). *Lpcat4* expression was undetectable in the liver.

Analysis of plasma lipid levels revealed lower plasma TG levels following an overnight fast in L-Lpcat3 KO mice compared to controls (*Figure 2C*). Levels of plasma total cholesterol and non-esterified free fatty acids (NEFA) were not different between groups. Body weight and fasting blood glucose levels were also not different between groups (*Figure 2—figure supplement 1*). Although total levels of plasma apolipoprotein B (apoB) were similar between groups (*Figure 2D*, *Figure 2—figure supplement 2B*), fractionation of plasma lipoproteins revealed lower levels of apoB in the VLDL fraction in L-Lpcat3 KO mice (*Figure 2E*, *Figure 2—figure supplement 2A*). Moreover, TG levels in the VLDL fraction were markedly reduced. We also observed a trend towards TG stores in the liver of L-Lpcat3 KO mice, along with histological evidence of increased lipid accumulation (*Figure 2F,G*).

As a complement to our analysis of L-Lpcat3 KO mice, which lack Lpcat3 expression in their livers from birth, we acutely deleted Lpcat3 in the liver of adult *Lpcat3*<sup>fl/fl</sup> mice with a Cre-expressing adenoviral vector. Interestingly, acute inactivation of Lpcat3 resulted in a more prominent decrease in fasting plasma TG levels compared to developmental deletion (*Figure 2H*). Furthermore, acute deletion uncovered a decrease in ad-lib plasma TG levels that was not observed with developmental

**Table 1**. Breeding data for global Lpcat3-deficient mice

| Genotype | Number of pups/mice | Observed % | Expected % | Time |
|----------|---------------------|------------|------------|------|
| WT | 25 | 28 | 25 | |
| Het | 43 | 48 | 50 | At birth |
| KO | 21 | 23 | 25 | |
| WT | 50 | 35 | 25 | |
| Het | 91 | 64 | 50 | At weaning |
| KO | 0 | 0 | 25 | |

Genotypic ratio of newborns and weanlings obtained from Lpcat3 heterozygote intercrosses.

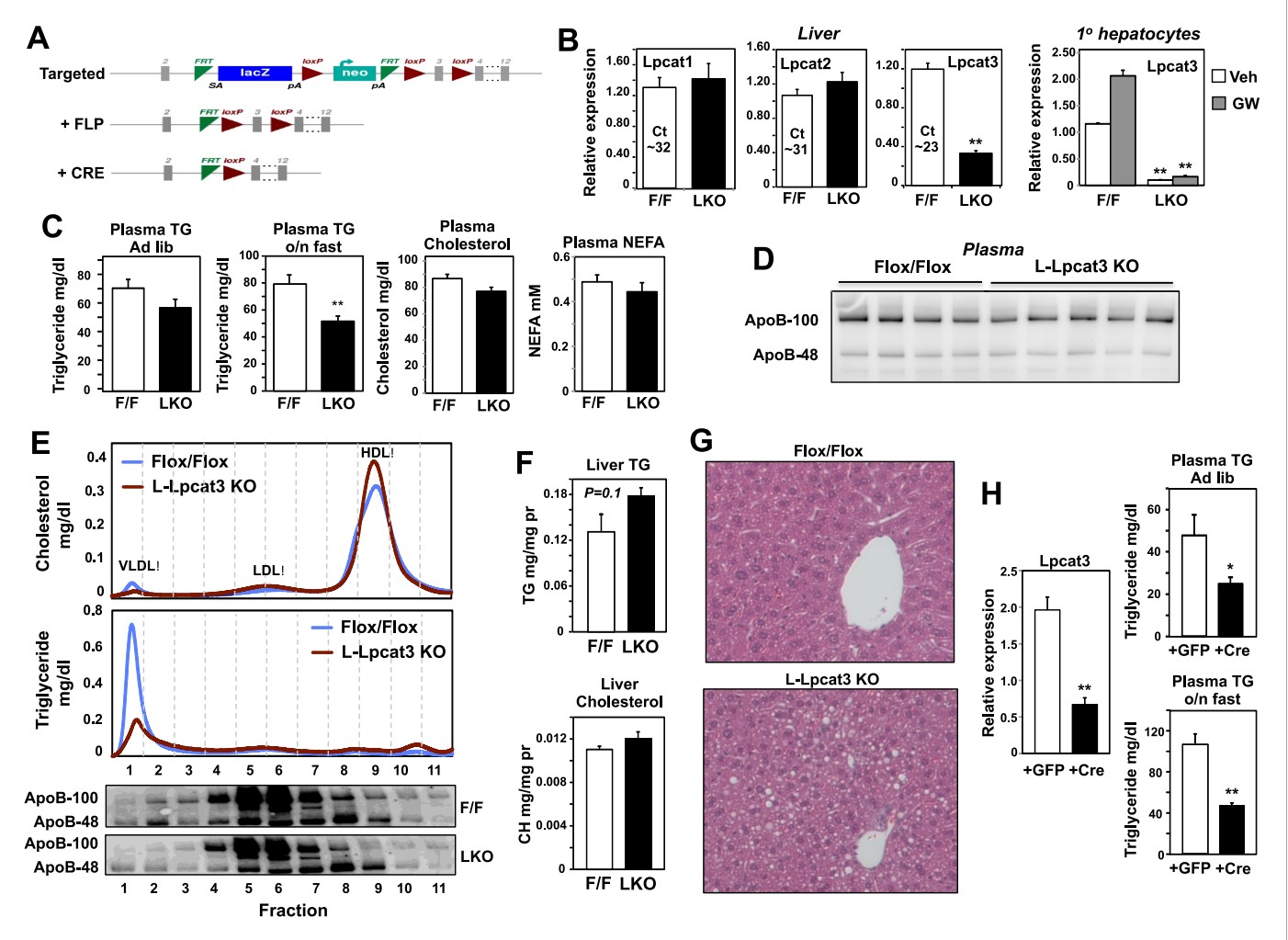

**Figure 2**. Altered triglyceride (TG) metabolism in liver-specific *Lpcat3* knockout mice. (**A**) Strategy for generating tissue-specific Lpcat knockout mice. Lpcat3⁻/⁺ mice carrying the conditional-ready knockout allele were mated with Flpe transgenic mice to generate the Lpcat3fl/fl mice. Lpcat3fl/fl mice were bred with tissue-specific Cre transgenic mice to generate tissue-specific Lpcat3 knockout mice. (**B**) Expression of Lpcat family members in liver of *Lpcat3^{fl/fl}* (F/F) and *Lpcat3^{fl/fl} Albumin-Cre* (L-KO) mice fed a chow diet. Gene expression was quantified by real-time PCR (n ≥ 6/group) (left panel). Primary hepatocytes from *Lpcat3^{fl/fl}* (F/F) and *Lpcat3^{fl/fl} Albumin-Cre* (L-KO) mice were treated overnight with 1 μM LXR agonist GW3965 (GW). Lpcat3 expression was quantified by real-time PCR (right panel). Ct values in *Lpcat3^{fl/fl}* liver samples were shown. Values are means ± SEM. (**C**) Plasma lipid levels in chow diet-fed *Lpcat3^{fl/fl}* (F/F) and *Lpcat3^{fl/fl} Albumin-Cre* (L-KO) mice under fed (*ad libitum*) or overnight (o/n) fasting (n ≥ 8/group). Plasma levels of cholesterol and NEFA were measured after an overnight fast. Values are means ± SEM. (**D**) Plasma was harvested from chow diet-fed *Lpcat3^{fl/fl}* (F/F) and *Lpcat3^{fl/fl} Albumin-Cre* (L-KO) mice after an overnight fast. ApoB protein was analyzed by western blotting and quantified (*Figure 2—figure supplement 2B*). (**E**) Plasma from *Lpcat3^{fl/fl}* (F/F) and *Lpcat3^{fl/fl} Albumin-Cre* (L-Lpcat3-KO) mice fasted overnight was pooled (n = 5). Lipoprotein profiles were analyzed by fast protein liquid chromatography (FPLC) (upper panel). ApoB protein in each corresponding fraction was analyzed by western blot (lower panel) and quantified (*Figure 2—figure supplement 2A*). (**F**) Lipid contents in livers of chow diet-fed *Lpcat3^{fl/fl}* (F/F) and *Lpcat3^{fl/fl} Albumin-Cre* (L-KO) mice fasted overnight (n ≥ 8/group). Values are means ± SEM. (**G**) Hematoxylin and eosin staining of liver sections from chow-fed *Lpcat3^{fl/fl}* (F/F) and *Lpcat3^{fl/fl} Albumin-Cre* (L-KO) mice after an overnight fast. (**H**) Liver expression of Lpcat3 (left panel) and plasma TG levels (right panel) in *Lpcat3^{fl/fl}* mice 3 weeks after being injected with an adenovirus encoding GFP or *Cre*. Mice were sacrificed under fed conditions (ad libitum) or after overnight fasting (o/n) (n ≥ 6/group). Gene expression was measured by real-time. Values are means ± SEM. Statistical analysis was performed with a Student's *t*-test (**B**, **C**, **F**, **H**). *p < 0.05; **p < 0.01.

The following figure supplements are available for figure 2:

**Figure supplement 1**. Blood glucose levels and body weight in control and liver-specific Lpcat3 KO mice (LKO).

**Figure supplement 2**. Quantification of western blots.

**Table 2**. Breeding data for liver-specific Lpcat3-deficient mice

| Genotype | Number of mice | Observed % | Expected % |
|---|---|---|---|
| Flox/Flox | 26 | 53 | 50 |
| Flox/Flox[albumin-cre] | 23 | 46 | 50 |

Genotypic ratio of adult mice obtained from $Lpcat3^{fl/fl}$ and $Lpcat3^{fl/fl}$ Albumin-Cre intercrosses.

deletion (*Figure 2C,H*). This finding suggests that there may be partial compensation for the chronic loss of Lpcat3 in the L-Lpcat3 KO mice. Collectively, the data of *Figure 2* are consistent with a potential role for Lpcat3 in hepatic TG metabolism.

To further explore a role for Lpcat3 in TG secretion, we challenged control and L-Lpcat3 KO mice with western diet (40% high fat and 0.2% cholesterol) for 9 weeks. Mice lacking hepatic Lpcat3 again showed lower total plasma TG levels (*Figure 3A*) and a striking loss of TG in the VLDL plasma lipoprotein fraction (*Figure 3B*, *Figure 3—figure supplement 1A*). In addition, these mice had increased levels of plasma apoB-100 compared to controls (*Figure 3C*, *Figure 3—figure supplement 1B*). Analysis of tissue lipids revealed prominent accumulation of hepatic TG and cholesterol in the liver of L-Lpcat3 KO mice (*Figure 3D,E*). We also challenged mice with a high-sucrose diet, which strongly stimulates hepatic lipid synthesis and secretion. On the high-sucrose diet, L-Lpcat3 KO mice exhibited hepatic TG accumulation (*Figure 3F,G*). The accumulation of hepatic TG in L-Lpcat3 KO mice in the setting of reduced plasma VLDL TG implied that Lpcat3 activity might be crucial for the assembly and secretion of TG-rich lipoproteins from the liver.

Lpcat3 is expressed at high levels in intestine as well as in the liver. We showed previously that hepatic *Lpcat3* expression is regulated by the sterol-activated nuclear receptor LXR (*Rong et al., 2013*). Here, we showed that intestinal Lpcat3 expression is strongly responsive to the administration of a synthetic LXR-agonist, GW3965 (*Figure 4A*). To address whether Lpcat3 activity may also be important for TG metabolism in intestinal enterocytes, we generated intestine-specific Lpcat3 KO mice (I-Lpcat3 KO) by crossing the floxed mice to villin-*Cre* transgenics. I-Lpcat3 KO mice were born at the predicted Mendelian frequency, and their body weights at birth were comparable to controls (*Table 3*, *Figure 4B*). However, even though the pups suckled, they failed to thrive and showed severe growth retardation by 1 week of age (*Figure 4C*). Expression of *Lpcat3* was reduced more than 90% in duodenum of I-Lpcat3 KO mice as expected, and there was no compensatory increase in expression of *Lpcat1*, *Lpcat2* or *Lpcat4* (*Figure 4D*). Blood glucose levels in 1-week-old I-Lpcat3 pups were very low (*Figure 4E*), consistent with results obtained with global knockouts (*Figure 1*). Plasma insulin levels were also correspondingly reduced. Plasma TG levels were lower and total cholesterol and NEFA levels were unchanged in I-Lpcat3 KO pups (*Figure 4E*). Histological analysis of intestines from I-Lpcat3 KO pups revealed a dramatic accumulation of cytosolic lipid droplets in intestinal enterocytes (*Figure 4F*), a phenotype reminiscent of intestinal apoB-deficient mice. Analysis of intestinal gene expression in I-Lpcat3 KO mice revealed reduced expression of several genes linked to intestinal TG metabolism, including *Apob*, *Cd36*, *Dgat2*, and *Mogat2* (*Figure 4G*). Given the massive enterocyte lipid accumulation in enterocytes, it is conceivable that some of those gene-expression changes were due, at least in part, to poor nutrition or cell toxicity. Nevertheless, these data were consistent with a role for Lpcat3 in TG mobilization and secretion–in the intestine as well as in the liver.

To gain insight into how the enzymatic activity of Lpcat3 was linked to these phenotypes, we performed lipidomic analyses. Previous studies using in vitro systems have profiled the substrate specificities of the four mammalian Lpcat family members (*Hishikawa et al., 2008*; *Zhao et al., 2008*). These studies suggested that Lpcat3 exhibited a preference for LysoPC and polyunsaturated fatty acyl-CoAs as substrates. Consistent with those results, we previously observed subtle changes in the levels of PC species containing polyunsaturated acyl in response to acute Lpcat3 knockdown in hepatocytes and macrophages (*Rong et al., 2013*). However, the consequences of a genetic inactivation of *Lpcat3* for membrane composition in vivo have never been addressed. Unexpectedly, analysis of phospholipid species in whole liver extracts from L-Lpcat3 KO mice by ESI-MS/MS revealed that Lpcat3 activity is uniquely required for the incorporation of arachidonate chains into PLs. Thus, despite the fact that Lpcat3 can catalyze the esterification of multiple unsaturated acyl chains into PC

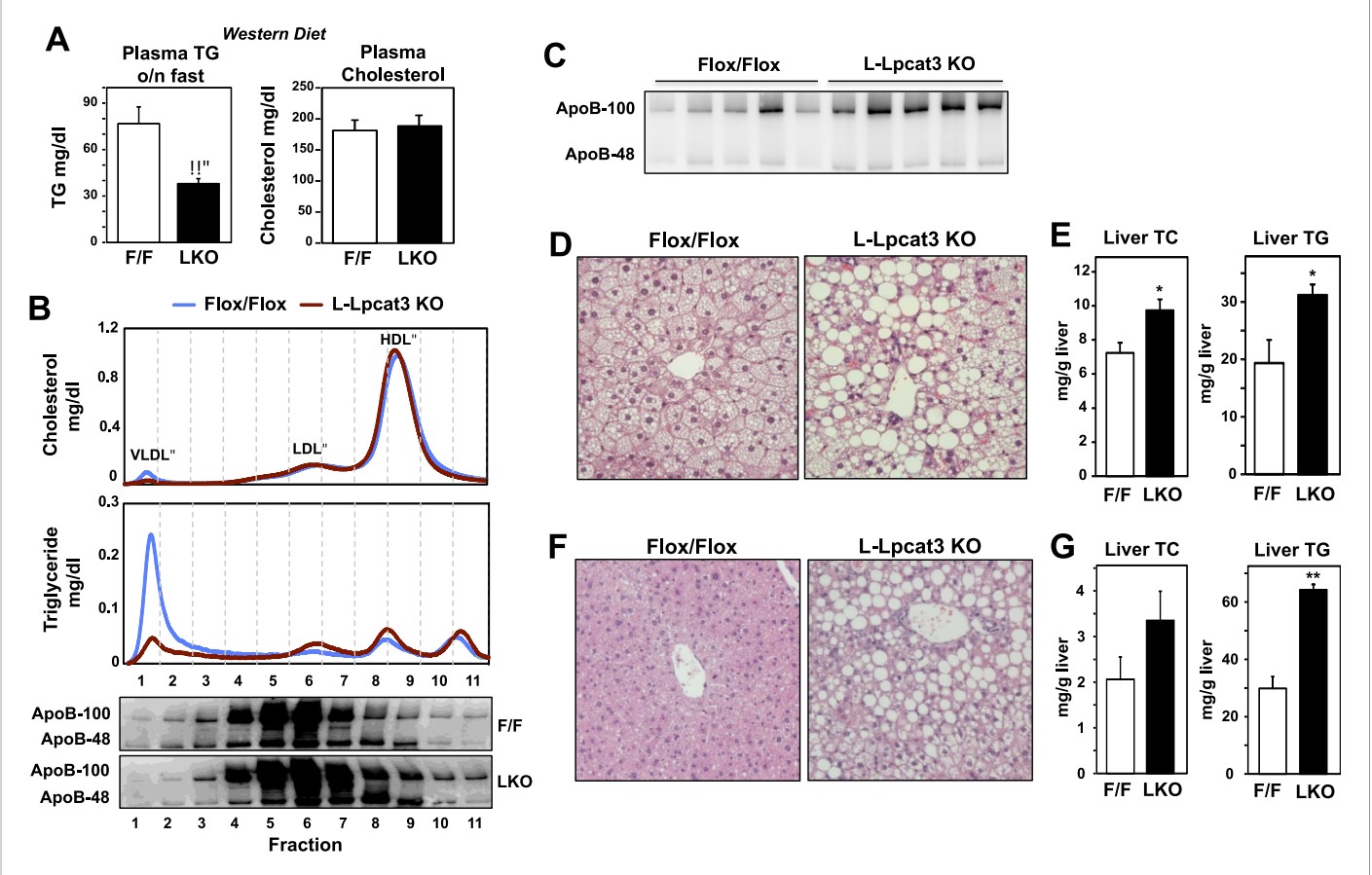

**Figure 3**. Dietary challenge accentuates metabolic phenotypes in liver-specific *Lpcat3* knockout mice. (**A**) Plasma lipids of *Lpcat3*^fl/fl (F/F) and *Lpcat3*^fl/fl *Albumin-Cre* (L-KO) mice fed on a western diet for 9 weeks. Plasma was collected from mice fasted for 6 hr ($n \geq 8$/group). Values are means ± SEM. (**B**) Plasma samples same as in (**A**) were pooled ($n = 5$). Lipoprotein profile was analyzed by FPLC (upper panel). ApoB protein in each fraction was analyzed by western blot (lower panel) and quantified (*Figure 3—figure supplement 1A*). (**C**) ApoB protein in plasma samples same as in (**A**) was analyzed by western blot and quantified (*Figure 3—figure supplement 1B*). (**D–E**) Hematoxylin and eosin staining (**D**) and lipid contents (**E**) of livers from western diet-fed *Lpcat3*^fl/fl (F/F) and *Lpcat3*^fl/fl *Albumin-Cre* (L-KO) mice. Values are means ± SEM. (**F–G**) Hematoxylin and eosin staining (**F**) and lipid contents (**G**) of livers from high sucrose diet-fed *Lpcat*3fl/fl (F/F) and Lpcat3fl/fl Albumin-Cre (L-KO) mice. Mice were fed on diet for 3 weeks and sacrificed after 6 hr fasting. Values are means ± SEM. Statistical analysis was performed using Student's *t*-test (**A**, **E** and **G**). *p < 0.05; **p < 0.01.

The following figure supplement is available for figure 3:

**Figure supplement 1**. Quantification of western blots.

in vitro, and despite the fact that Lpcat3 is by far the most abundant Lpcat family member in the liver, the consequences of loss of Lpcat3 for PL composition were remarkably selective. The total level of PC was not different between L-Lpcat3 KO and control livers (*Figure 5A*). However, there were striking reductions (~70%) in the abundance of 16:0, 20:4 PC and 18:0, 20:4 PC (nomenclature: a:b, c:d; where a and c are the number of carbons and b and d are the number of double bonds of the sn-1 and sn-2 aliphatic groups, respectively), two of the most abundant arachidonoyl PC species in liver membranes (*Figure 5A*). However, the actual reduction of 16:0, 20:4 PC and 18:0, 20:4 PCs in *hepatocyte* membranes was almost certainly much greater, given that a significant fraction of membranes in total liver extracts originate from other cell types (where Lpcat3 activity is preserved). Furthermore, we observed compensatory increases in the abundance of other PC species in L-Lpcat3 KO livers, notably those containing monounsaturated chains (e.g., 16:0, 18:1 PC and 18:0, 18:1 PC; *Figure 5A*).

We also analyzed the effect of western diet on hepatic phospholipid composition in the presence and absence of Lpcat3. The diet increased the abundance of certain PC species, such as 16:0, 18:1 PC,

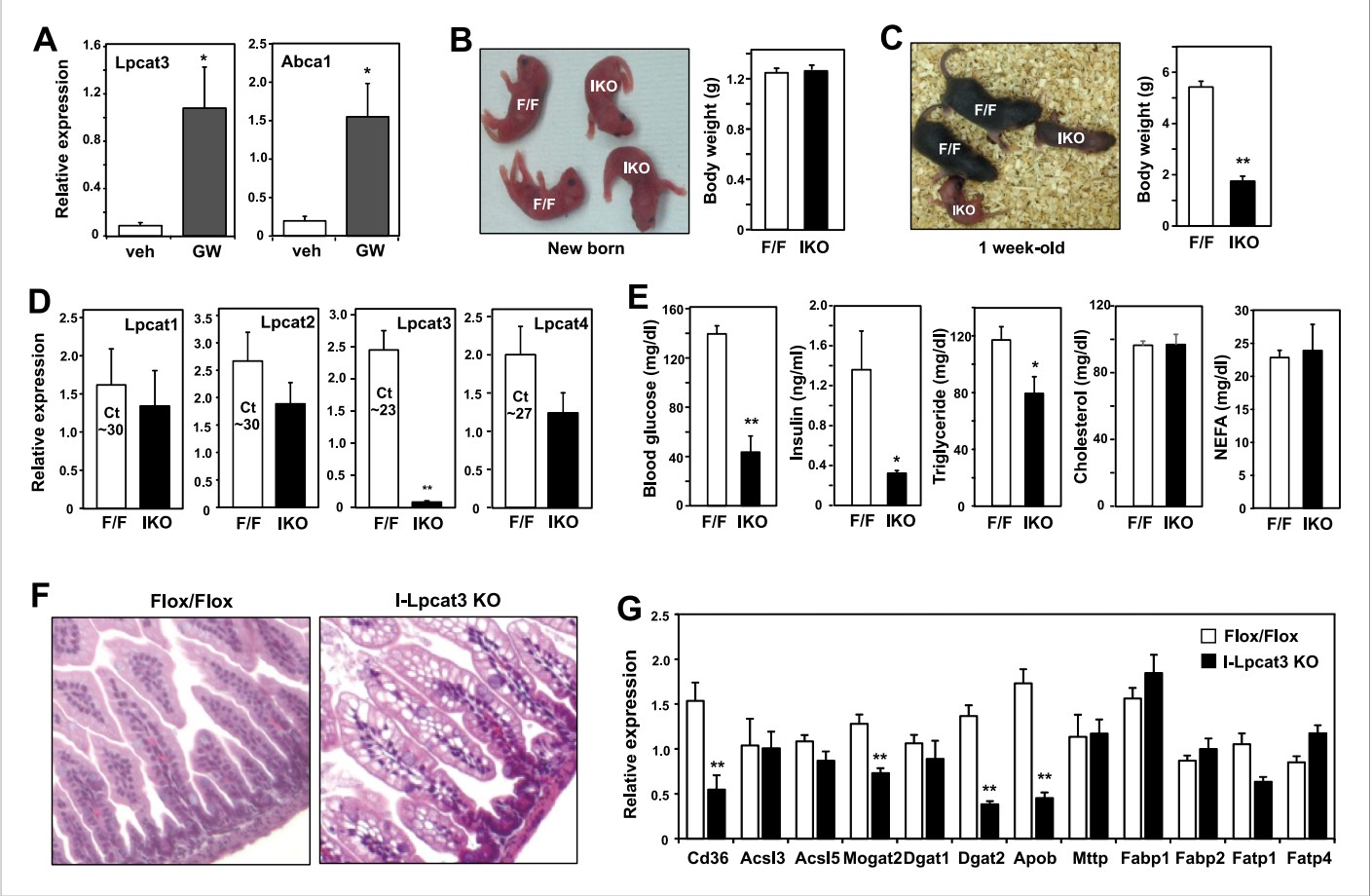

**Figure 4.** Altered TG metabolism in intestine-specific Lpcat3 knockout mice. (**A**) Induction of Lpcat3 mRNA expression in duodenum of mice treated with 40 mg/kg/day GW3956 by oral gavage for 3 days (n = 5/group). Gene expression was measured by real-time PCR. Values are means ± SEM. (**B**) Representative photograph and body weight of newborn *Lpcat3^{fl/fl} Villin-cre* (IKO) and control *Lpcat3^{fl/fl}* (F/F) pups (n = 5/group for body weight measurement). Values are means ± SEM. (**C**) Representative photograph and body weight of 1 week-old *Lpcat3^{fl/fl} Villin-cre* (IKO) and control *Lpcat3^{fl/fl}* (F/F) pups (n ≥ 6/group for body weight measurement). Values are means ± SEM. (**D**) Expression of Lpcat family members in 1 week-old *Lpcat3^{fl/fl}* (F/F) and *Lpcat3^{fl/fl} Villin-cre* (IKO) duodenum measured by real-time PCR (n ≥ 6/group). Ct values of F/F samples were shown. Values are means ± SEM. (**E**) Blood glucose, plasma lipids and insulin levels in 1 week-old *Lpcat3^{fl/fl}* (F/F) and *Lpcat3^{fl/fl} Villin-cre* (IKO) pups (n ≥ 6/group). Values are means ± SEM. (**F**) Hematoxylin and eosin staining of intestines from 1 week-old *Lpcat3^{fl/fl}* (WT) and *Lpcat3^{fl/fl} Villin-cre* (IKO) pups. (**G**) Expression of genes in duodenum of 1 week-old *Lpcat3^{fl/fl}* (WT) and *Lpcat3^{fl/fl} Villin-cre* (IKO) pups. Gene expression was measured by real-time PCR (n ≥ 6/group). Values are means ± SEM. Statistical analysis was performed using Student's t-test (**A**, **B**, **D**, **E** and **F**). *p < 0.05; **p < 0.01.

presumably reflecting the abundance of oleate in the western diet. However, we observed the same prominent deficits in 16:0, 20:4 PC and 18:0, 20:4 PC on western diet as we observed on chow diet, with no change in total levels of PC (*Figure 5B*). In addition, there were reductions in 16:1, 18:2 PC and 18:1, 20:4 PC in western diet-fed Lpcat3 KO livers compared to controls. Severe reductions of phosphatidlyethanolamine (PE) species containing arachidonate chains were also observed in L-Lpcat3 KO mice on both chow and western diets (*Figure 5—figure supplement 1*). Interestingly, 16:0, 20:4 PE and 18:0, 20:4 PE are particularly abundant PE species in liver on western diet, and reductions in their levels was sufficient to reduce total PE levels.

Since unsaturated PC species are abundant in plasma lipoproteins, and since mice lacking Lpcat3 in liver or intestine show reduced plasma TG levels, we tested whether loss of Lpcat3 expression in liver would alter the phospholipid composition of VLDL secreted from the liver. After fasting L-Lpcat3 KO and control mice overnight, the VLDL was isolated from pooled plasma (n = 5 mice/group). Analysis of PC species in the pooled VLDL fractions by ESI-MS/MS revealed highly selective reductions in 16:0, 20:4 PC and 18:0, 20:4 PC (*Figure 5C*). Thus, the phospholipid deficits of L-Lpcat3 KO hepatic

**Table 3.** Breeding data for intestine-specific Lpcat3-deficient mice

| Genotype | Number of pups/mice | Observed % | Expected % | Time |
|---|---|---|---|---|
| WT | 25 | 28 | 25 | |
| Het | 43 | 48 | 50 | At birth |
| KO | 21 | 23 | 25 | |

| Genotype | Number of pups | Observed % | Expected % | Time |
|---|---|---|---|---|
| F/F, Cre+ | 20 | 24 | 25 | |
| F/F, Cre− | 24 | 28 | 25 | At birth |
| F/+, Cre+ | 19 | 22 | 25 | |
| F/+, Cre− | 20 | 24 | 25 | |
| F/F, Cre+ | 12 | 16 | 25 | |
| F/F, Cre− | 24 | 32 | 25 | 1 week old |
| F/+, Cre+ | 19 | 25 | 25 | |
| F/+, Cre− | 20 | 26 | 25 | |

Genotypic ratio of newborns and 1 week-old pups obtained from *Lpcat3*<sup>fl/fl</sup> and *Lpcat3*<sup>+/F</sup> *Villin-Cre* intercrosses.

membranes are passed on to the VLDL particles that they generate, strongly suggesting that loss of these PC species is mechanistically related to the altered TG content of the particles.

Interestingly, the severe loss of arachidonate in livers of Lpcat3 KO mice on chow diet was observed in PLs, but not in TG (*Figure 5D*) nor in cholesterol esters (*Figure 6A*). There was an increase in total cholesterol ester and a number of cholesterol ester species in L-Lpcat3 KO mice, consistent with the histologic evidence of increased neutral lipid content in the liver (*Figure 2*). There was no accumulation of the major lipid substrates of Lpcat3 (16:0 lysoPC and 18:0 lysoPC) in L-Lpcat3 KO mice (*Figure 6B*), suggesting these precursors are efficiently shuttled into alternative pathways in the absence of Lpcat3. Broadly similar results were obtained from mice fed western diet, although we did observe modestly lower levels of 20:4 lysoPC (*Figure 6C,D*).

To determine the consequence of loss of Lpcat3 activity for hepatic gene expression we performed transcriptional profiling. Despite the marked changes in membrane composition in L-Lpcat3 KO livers, the effect on gene expression was surprisingly limited (*Figure 7A*). On chow diet, only a handful of genes were altered more than twofold. Among the highest changes were increased expression of the lipoprotein remodeling enzyme *Pltp* and the lipid transport protein *Cd36*. A number of additional genes involved in lipid metabolism were induced to a more modest degree, including *Txnip* and *Fabp2*. Interestingly, many of the genes that were induced are known targets for the lipid-activated nuclear receptor PPARα, suggesting a compensatory gene expression response to increased cellular lipid levels in the absence of Lpcat3 (*Figures 2, 3*).

More prominent changes in gene expression in L-Lpcat3 KO mice were observed in the setting of western diet feeding (*Figure 7B*). Multiple genes involved in lipid metabolism and transport and in lipid droplet formation were upregulated, again likely reflecting a response to increased hepatic TG and cholesterol content. Interestingly, the putative transporter Mfsd2a (*Nguyen et al., 2014*) was induced in Lpcat3-deficient livers on both chow and western diet, perhaps as compensation for the loss of arachidonoyl PLs. In addition, there was increased expression of a number of genes linked to inflammation and inflammatory cell recruitment (*Figure 7C*). These data are consistent with our earlier study showing that adenoviral expression of Lpcat3 is protective against hepatic inflammation in the setting of lipid excess (*Rong et al., 2013*). We validated the microarray results for a number of genes by real-time PCR and also assessed the expression of other genes relevant to lipoprotein production (*Figure 7D,E*). Importantly, there was no change in the expression of mRNAs encoding apoB or of the critical lipid transfer protein MTTP in livers of Lpcat3 KO mice, indicating that altered plasma TG levels cannot be attributed to changes in expression of these factors.

We previously reported that acute knockdown of Lpcat3 expression in livers of genetically obese mice exacerbated lipid-induced ER stress (*Rong et al., 2013*). Genetic deletion of Lpcat3 from liver

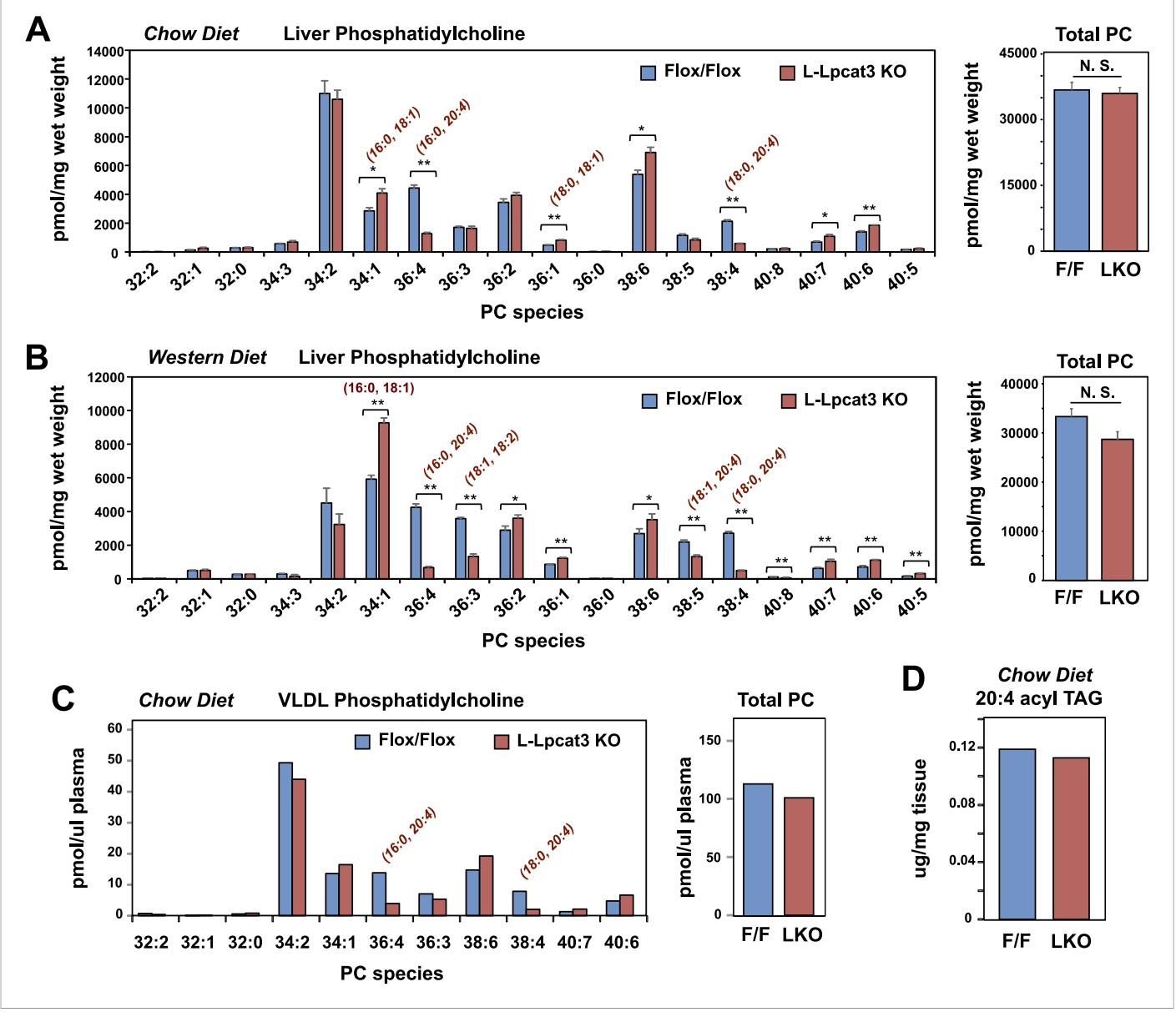

**Figure 5**. Lpcat3 is required for the incorporation of arachidonate into phosphatidylcholine (PC) in mouse liver and very low-density lipoprotein (VLDL). (**A–B**) ESI-MS/MS analysis of the abundance of PC species in livers from Lpcat3fl/fl (Flox/Flox) and Lpcat3fl/fl Albumin-Cre (L-Lpcat3 KO) mice fed on a chow diet (**A**) and a western diet (**B**) (n ≥ 5/group). (**C**) ESI-MS/MS analysis of the abundance of PC species in plasma VLDL fraction from Lpcat3fl/fl (fl/fl) and Lpcat3fl/fl Albumin-Cre (L-Lpcat3 KO) mice fed on a chow diet. Plasma was harvested from mice after overnight fasting. VLDL fractions were pooled from 5 mice/group. (**D**) GC-FID analysis of the abundance of arachidonoyl acyl chain in triglyceride (TAG) in livers from Lpcat3fl/fl (F/F) and Lpcat3fl/fl Albumin-Cre (L-KO) mice fed on a chow diet. Statistical analysis was performed using Student's *t*-test. Values are means ± SEM. *p < 0.05; **p < 0.01.

The following figure supplement is available for figure 5:

**Figure supplement 1**. Lpcat3 is required for the incorporation of arachidonate into phosphatidylethanolamine in mouse liver.

did not lead to increased mRNA expression of ER stress markers in mice fed chow or western diet (*Figure 7—figure supplement 1*). These observations suggest that there may be compensatory responses in membrane composition that prevent induction of the ER stress response in the setting of chronic Lpcat3 deletion. In support of this idea, we observed a prominent increase in the abundance of oleoyl-PC species in L-Lpcat3 mice (*Figure 5*). We cannot exclude the possibility that ER stress may be increased in L-Lpcat3 mice in the setting of genetic obesity or other causes of severe lipotoxicity.

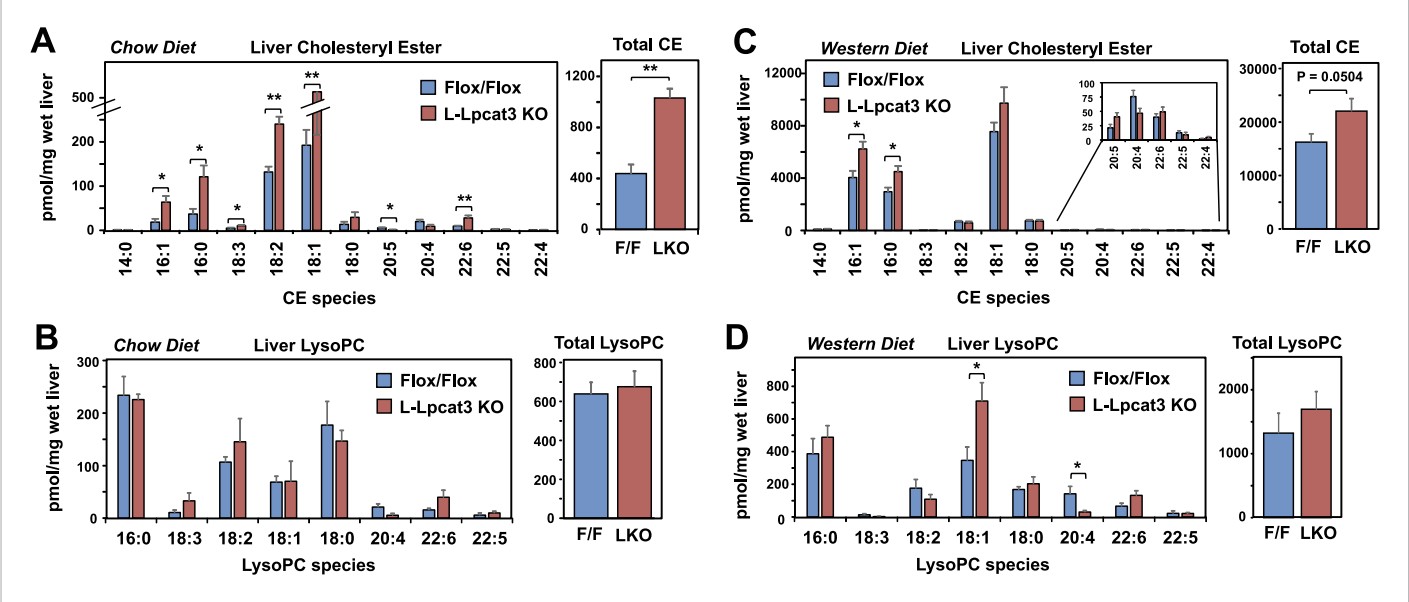

**Figure 6**. Analysis of cholesteryl ester and lysoPC species in L-Lpcat3 KO mice. (**A–B**) ESI-MS/MS analysis of the abundance of cholesteryl ester (**A**) and lysophosphatidylcholine (LysoPC) (**B**) species in livers of *Lpcat3^{fl/fl}* (Flox/Flox) and *Lpcat3^{fl/fl} Albumin-Cre* (L-Lpcat3 KO) mice fed on a chow diet (*n* ≥ 5/group). (**C–D**) ESI-MS/MS analysis of the abundance of cholesteryl ester (**C**) and lysophosphatidylcholine (LysoPC) (**D**) species in livers of *Lpcat3^{fl/fl}* (Flox/Flox) and *Lpcat3^{fl/fl} Albumin-Cre* (L-Lpcat3 KO) mice fed on a western diet (*n* ≥ 5/group). Statistical analysis was performed using Student's *t*-test. Values are means ± SEM. *p < 0.05; **p < 0.01.

We next investigated the etiology of reduced VLDL TG levels in mice lacking Lpcat3. Protein levels of apoB in liver were not different between control and L-Lpcat3 KO mice (*Figure 8A*). Together with the preserved levels of apoB in plasma (*Figures 2, 3*), these data suggest that a defect in production or secretion of apoB alone cannot explain the change in plasma VLDL. We directly assessed hepatic TG secretion by treating mice with the lipase inhibitor Tyloxapol and measuring TG accumulation in fasted mice. L-Lpact3 KO mice showed a markedly reduced rate of TG secretion (*Figure 8B*), suggesting that reduced incorporation of TGs into lipoproteins was the likely cause of the reduced VLDL TG levels. Consistent with this hypothesis, negative staining of plasma VLDL fractions by EM revealed markedly smaller VLDL particles in L-Lpcat3 KO mice compared to controls (*Figure 8C*).

To obtain further insight into the nature of the lipoprotein production defect in L-Lpcat3 KO mice, we examined liver samples by EM. Nascent lipoproteins, ranging between 0.05 and 0.11 microns in diameter, were easily visualized in the Golgi apparatus and secretory vesicles of control mice (*Figure 9*). Lipoprotein particles were also present in Golgi and secretory vesicles of L-Lpcat3 KO mice, however they were markedly smaller, ranging between 0.03 to 0.08 microns in diameter (*Figure 9*). We also observed small lipoprotein particles in the ER in mice of both genotypes (*Figure 9—figure supplement 1*). We did not find differences in the morphology of Golgi, ER or mitochondria between control and L-Lpcat3 KO livers, suggesting that loss of arachidonoyl PLs does not dramatically alter membrane structure in these organelles (*Figure 9—figure supplement 1*).

To further investigate the hypothesis that defective apoB lipidation was responsible for the phenotype of L-Lpcat3 KO mice, we isolated the Golgi apparatus from livers, fractionated the luminal contents by density gradient centrifugation, and analyzed the distribution of ApoB in the different fractions. There was reduced apoB in the most buoyant lipoprotein fractions, consistent with reduced lipidation of apoB particles in the absence of Lpcat3 (*Figure 10A*). In line with this finding, we found reduced TG levels in Golgi membrane fractions isolated by density gradient centrifugation from L-Lpcat3 KO mice compared to controls in two independent purifications (*Figure 10B*).

To understand how changes in membrane phospholipid composition in Lpcat3-deficient mice might lead to reduced mobilization and secretion of TGs, we performed biophysical studies on the impact of Lpcat3 deficiency on lipid movement in living cells. Prior studies have employed the

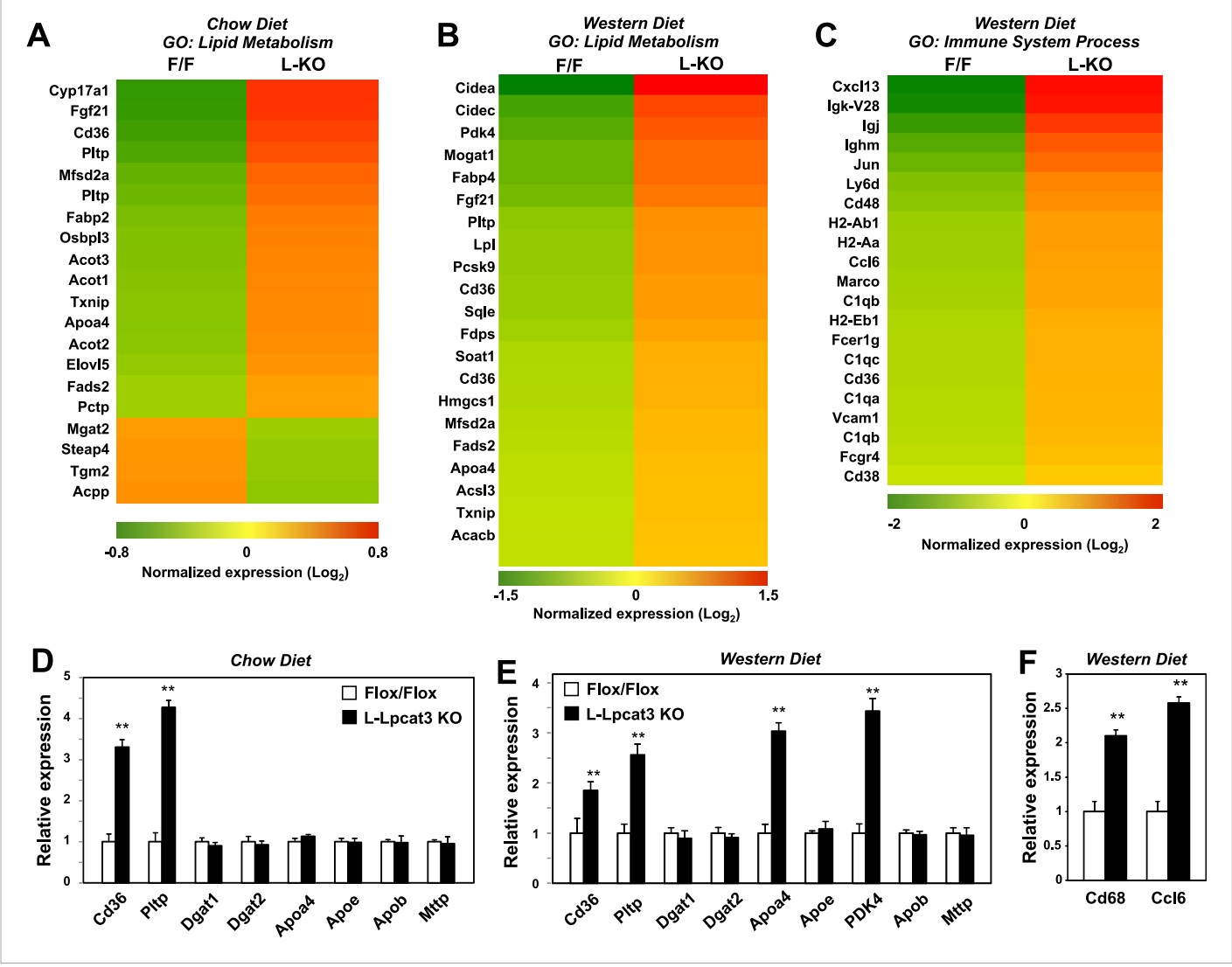

**Figure 7**. Altered gene expression linked to lipid metabolism and inflammation in liver-specific Lpcat3 knockout mice. (**A**) Gene expression in livers of *Lpcat3*$^{fl/fl}$ (F/F) and *Lpcat3*$^{fl/fl}$ *Albumin-Cre* (L-KO) mice fed on a chow diet was analyzed by Affymetrix arrays. Select genes under the gene ontology (GO) term 'lipid metabolism' are presented by heatmap. Samples from 5 mice/group were pool for analysis. (**B–C**) Gene expression in livers of *Lpcat3*$^{fl/fl}$ (F/F) and *Lpcat3*$^{fl/fl}$ *Albumin-Cre* (L-KO) mice fed on a western diet for 9 weeks was analyzed by Affymetrix arrays. Select genes under the GO term 'lipid metabolism' (**B**) and 'immune system process' (**C**) are presented by heatmap. Samples from 5 mice/group were pool for analysis. (**D**) Expression of selective lipid metabolism genes in livers of *Lpcat3*$^{fl/fl}$ (Fl/Fl) and *Lpcat3*$^{fl/fl}$ *Albumin-Cre* (L-Lpcat3 KO) mice fed on a chow diet was analyzed by real-time PCR (*n* ≥ 5/group). Values are means ± SEM. (**E–F**) Expression of selective lipid metabolism (**E**) and inflammation (**F**) genes in livers of *Lpcat3*$^{fl/fl}$ (Fl/Fl) and *Lpcat3*$^{fl/fl}$ *Albumin-Cre* (L-Lpcat3 KO) mice fed on a western diet was analyzed by real-time PCR (*n* ≥ 5/group). Values are means ± SEM. Statistical analysis was performed using Student's *t*-test (**D**, **E** and **F**). **p < 0.01.

The following figure supplement is available for figure 7:

**Figure supplement 1**. Chronic deletion of Lpcat3 does not induce unfolded protein response downstream gene expression.

lipophilic dye laurdan to interrogate lipid dynamics in membranes (*Parasassi and Gratton, 1995*; *Vest et al., 2006*; *Golfetto et al., 2013*). Laurdan is a fluorescent lipophilic molecule that can be used to detect changes in membrane dynamics due to its sensitivity to the polarity of the membrane environment. Changes in membrane dynamics shift the laurdan emission spectrum, which can be quantified by the generalized polarization (GP) calculated from the spectrum shifts (*Parasassi et al., 1990*). We isolated primary hepatocytes from L-Lpcat3 KO mice and controls, stained with laurdan,

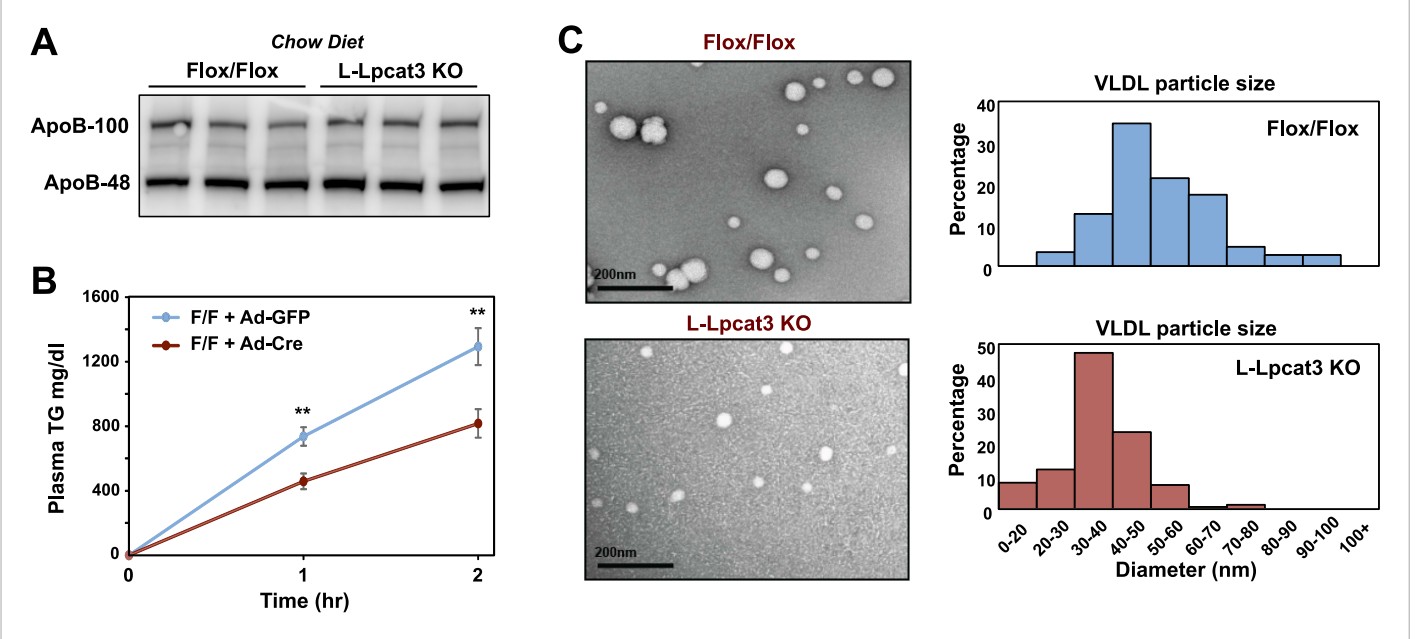

**Figure 8**. Loss of Lpcat3 from liver impairs hepatic TG secretion. (**A**) ApoB protein in livers from *Lpcat3*<sup>fl/fl</sup> (Flox/Flox) and *Lpcat3*<sup>fl/fl</sup> *Albumin-Cre* (L-Lpcat3 KO) mice fed on a chow diet was analyzed by western blot (n = 3). (**B**) VLDL-TG secretion in *Lpcat3*<sup>fl/fl</sup> mice transduced with adenoviral expressed Cre (Ad-Cre), compared to control GPF (Ad-GFP) for 4 weeks. Mice were fasted for 6 hr followed by intravenous injection of tyloxapol. Plasma TG was measured at indicated durations after tyloxapol injection. Values are means ± SEM. (**C**) Plasma VLDL particle size in *Lpcat3*<sup>fl/fl</sup> *Albumin-Cre* mice. Isolated VLDL particles from *Lpcat3*<sup>fl/fl</sup> (Fl/Fl) and *Lpcat3*<sup>fl/fl</sup> *Albumin-Cre* (L-Lpcat3 KO) mice fed on a western diet were stained with 2.0% uranyl acetate and visualized by electron microscopy (EM) (left) and their size quantified (right). Statistical analysis was performed using two-way ANOVA with Bonferroni post hoc tests (**B**). **$p < 0.01$.

and monitored fluorescence in live cells by microscopy. A visual representation of the calculated GP intensity is presented in *Figure 10C*, and the data are quantitated in *Figure 10D*. The prominent increase in GP (as shown by the yellow/orange pseudocolor signal) in cells lacking Lpcat3 is indicative of areas of cellular membrane with reduced dynamics. These data show that membranes are less dynamic, and their component lipids move less readily, in the absence of Lpcat3, an observation that is consistent with the reduced amount of arachidonate groups in membrane PLs. We propose that efficient transfer of bulk TG to nascent VLDL particles is enabled by the presence of arachidonoyl PLs in the ER/Golgi membrane due to its unique ability to facilitate optimal lipid movement and membrane dynamics.

Finally, as LXR agonists have previously been reported to promote hepatic TG secretion (*Schultz et al., 2000*), we tested the requirement for Lpcat3 induction in this effect. Control and L-Lpcat3 KO mice were treated for 2 days with the LXR agonist GW3965 (20 mpk/day) by oral gavage. The TG levels in fractionated plasma were then determined after 6 hr fast. In line with prior studies, treatment of control mice with LXR agonist led to a prominent (95%) increase in plasma VLDL TG levels (*Figure 11*). By contrast, treatment of L-Lpcat3 KO mice with LXR agonist had a much more modest effect (45% increase). These data suggest that induction of Lpcat3 expression plays an important role in the control of hepatic VLDL production by LXRs.

## Discussion

It has long been appreciated that phospholipid availability can influence the production of lipoproteins (*Vance, 2008*; *Abumrad and Davidson, 2012*), but the molecular basis of these connections, beyond the obvious need for sufficient PLs to coat the surface of lipoprotein particles, has been unclear. We have shown here that Lpcat3 is uniquely required for the incorporation of arachidonic acid into membranes, and that selective reduction in the abundance of these lipids impairs TG mobilization and lipoprotein production. These studies highlight a previously un-recognized requirement for a specific membrane lipid class in lipoprotein metabolism.

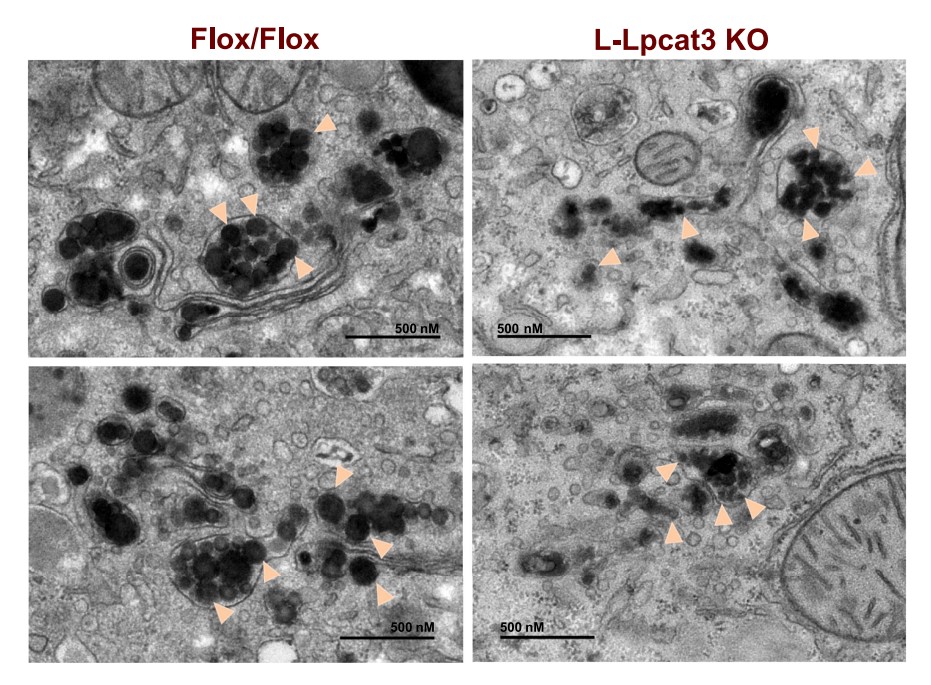

**Figure 9**. Reduced nascent lipoprotein particle size in the lumen of Golgi and secretory vesicles in Lpcat3-deficient liver. EM of imidazole-stained liver sections. *Lpcat3*fl/fl (Fl/Fl) and *Lpcat3*fl/fl *Albumin-Cre* (L-Lpcat3 KO) mice were fasted for 6 hr. Samples were fixed and processed as described in the methods. Arrowheads indicate nascent lipoprotein particles.

The following figure supplement is available for figure 9:

**Figure supplement 1**. Loss of Lpcat3 in liver does not alter membrane structure in ER and mitochondria.

Earlier studies using in vitro systems have demonstrated that changes in polyunsaturated PL can affect membrane-associated biological processes, including the assembly of signalsomes on membranes and endocytosis (*Koeberle et al., 2013*; *Pinot et al., 2014*). However, assessing the physiological impact of changing the abundance of individual PL species in animals has been a difficult experimental problem. Our study identifies Lpcat3 as a critical determinant of the abundance of arachidonoyl PLs in mice. The largely preserved levels of other PL species containing polyunsaturated chains in Lpcat3-deficient livers suggests that other Lpcats may be able to catalyze the incorporation of certain polyunsaturated fatty acids into membranes in the absence of Lpcat3. However, Lpcat3 is apparently unique in its ability to catalyze arachidonoyl PL synthesis. Lpcat3-deficient mouse models therefore provide an unprecedented opportunity to study the physiological and pathophysiological consequences of manipulation of membrane phospholipid composition in vivo.

Previous research has elucidated a requirement for de novo PC biogenesis in the production and secretion of VLDL from liver. Feeding mice a choline-deficient diet or deleting genes involved in de novo PC synthesis (e.g., *Pemt*, *CT-α*) impairs VLDL secretion and induces hepatosteatosis (*Noga and Vance, 2003*; *Jacobs et al., 2004*). These models exhibit reduced total apoB protein in plasma as well as lower plasma TGs, suggesting that adequate PC biosynthesis is important for both apoB secretion and VLDL lipidation. In our study, mice lacking Lpcat3 retain the ability to secrete apoB and are capable of producing small poorly-lipidated VLDL particles. However, they are unable to transfer TG to lipoproteins at an appropriate rate in the setting of increased metabolic demand. Our results suggest that the reduced availability of arachidonoyl PLs becomes especially problematic when there is a need to mobilize a large bolus of TG into lipoproteins. For example, when L-Lpcat3 KO mice are challenged with the lipogenic sucrose diet, they are unable to efficiently mobilize the newly synthesized TG into plasma lipoproteins and it instead accumulates in cytosolic lipid droplets in the liver.

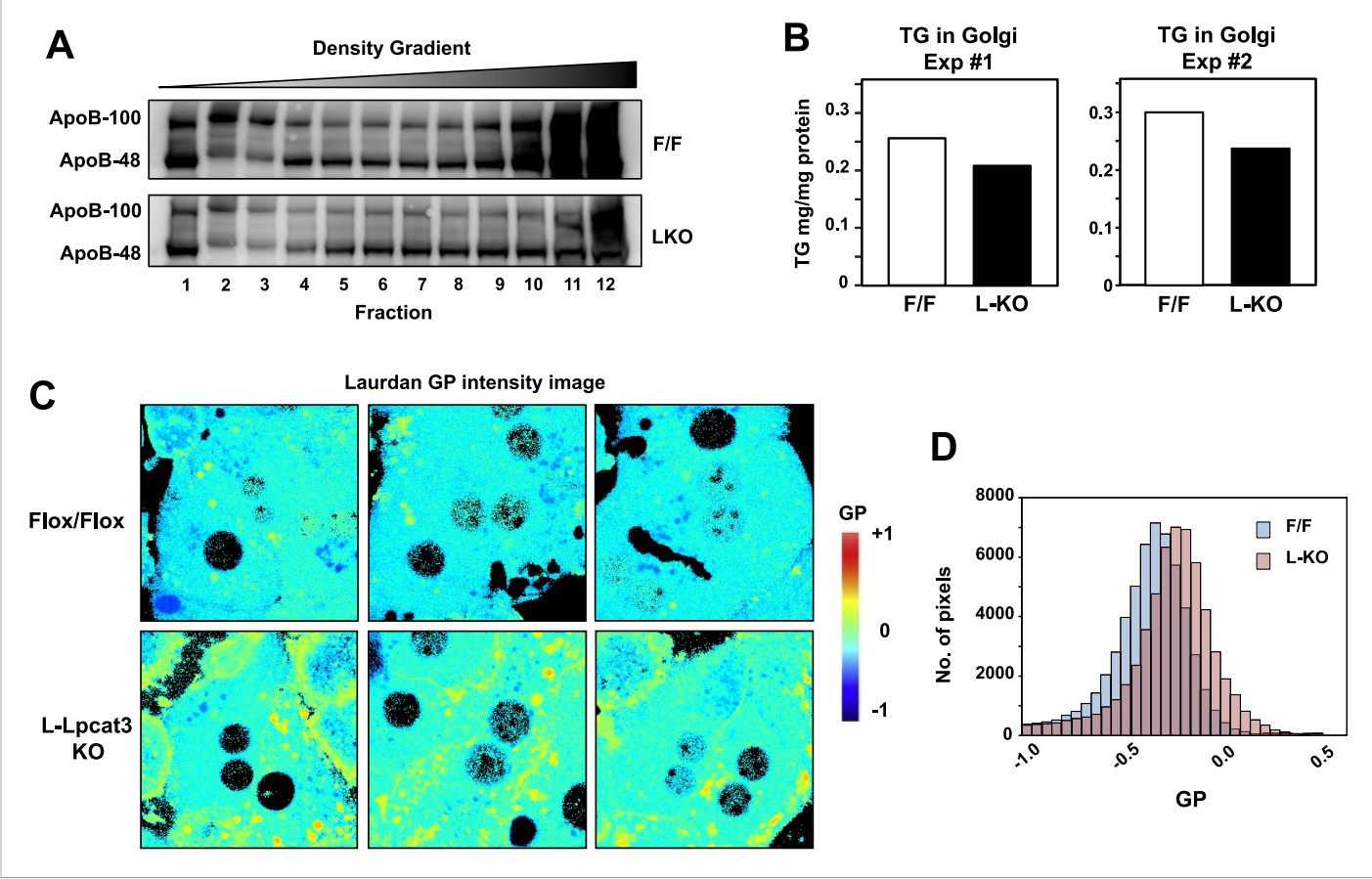

**Figure 10**. Lpcat3 activity regulates membrane lipid mobility and apoB lipidation in hepatocytes. (**A**) The distribution of apoB-containing lipoproteins in the Golgi fractions isolated from livers of *Lpcat3$^{fl/fl}$* (F/F) and *Lpcat3$^{fl/fl}$ Albumin-Cre* (LKO) mice. Golgi luminal contents were subject to a density gradient ultracentrifugation. 12 fractions from the gradient were harvested from top to bottom and analyzed by western blot. (**B**) Triglyceride contents of the Golgi fractions isolated from livers of *Lpcat3$^{fl/fl}$* (F/F) and *Lpcat3$^{fl/fl}$ Albumin-Cre* (LKO) mice. Results from two representative experiments were shown. (**C**) Live primary hepatocytes from *Lpcat3$^{fl/fl}$* (Flox/Flox) and *Lpcat3$^{fl/fl}$ Albumin-Cre* (L-Lpcat3 KO) mice were stained with laurdan. The laurdan emission spectrum was captured by a 2-photon laser-scan microscope. Generalized polarization (GP) was calculated from the emission intensities obtained from images. Higher GP value indicates that membranes are more ordered and less dynamic. The GP value of each pixel was used to generate a pseudocolor GP image. (**D**) The binary histograms of the GP distribution of the GP images ($n = 4$). The size of the GP binary is 0.05.

Similarly, mice lacking Lpcat3 in the intestine are unable to handle the high TG load of milk during suckling and accumulate large amounts of lipid in enterocytes. Given that Lpcat3 is the major Lpcat enzyme in enterocytes (*Figure 4D*), loss of Lpcat3 would be expected to impair the re-esterification of LysoPC to PC in enterocytes, which has been demonstrated to be important for lipid absorption. Unfortunately, the early lethality of the I-Lpcat3 KO pups presents a challenge to the analysis of lipid absorption. An inducible knockout system may prove a better model for studying the impact of Lpcat3 on intestinal lipid transport in adult mice.

Lpcat3 is a integral membrane protein of the ER and is therefore ideally positioned to produce arachidonoyl PC at the site of lipoprotein biogenesis (*Fisher and Ginsberg, 2002*; *Zhao et al., 2008*). Our observations suggest that Lpcat3 and its lipid products are likely not essential for the cotranslational lipidation of apoB and the generation of primordial VLDL particles (*Fisher and Ginsberg, 2002*). However, the small size of plasma VLDL particles, together with the reduced TG-rich apoB-containing particles in the Golgi fraction of L-Lpcat3 KO livers, strongly suggest that Lpcat3 impacts the second step of VLDL assembly–bulk TG addition to lipid-poor apoB particles and the generation of mature VLDL. TG-rich VLDL assembly is highly dependent on the efficient trafficking of stored cytosolic lipid from lipid droplets or newly synthesized lipid to primordial VLDL particles. One factor that could affect this transfer is the membrane environment where the bulk lipidation occurs.

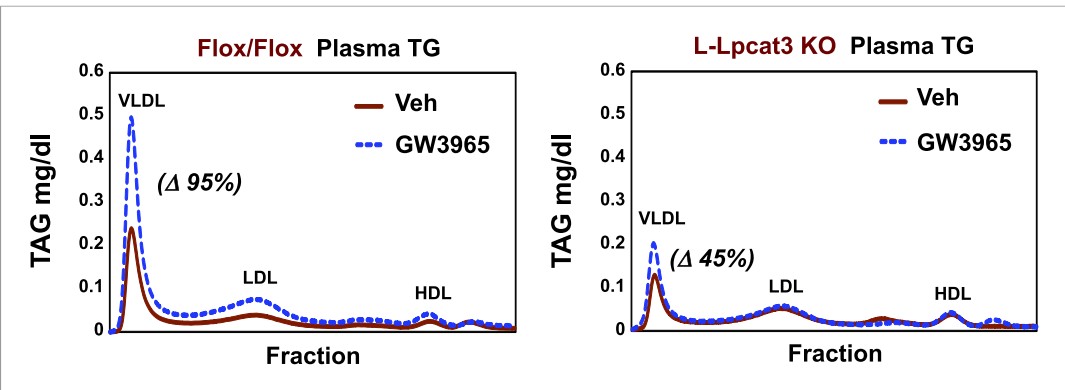

**Figure 11**. Lpcat3 is required for LXR-dependent VLDL TG secretion from liver. Plasma lipoprotein profile from flox/flox and L-Lpcat3 mice gavaged with LXR agonist GW 3965 (20 mg/kg) for 2 days. Pooled plasma (5 mice/group) was analyzed by FPLC. Increases in VLDL TG were calculated based on the area under the curve.

Although the precise mechanism that links membrane PL composition and VLDL lipidation is not clear, biophysical studies suggested that greater lipid transport is generally observed with more fluid and highly curved membrane surfaces (*Lev, 2012*).

Results from our laurdan staining experiments support the notion that the presence of arachidonoyl PLs in intracellular membranes promotes the dynamics of the membranes. However, detailing these biophysical changes and elucidating how they influence lipid movement will require further study with specialized tools and systems. Nevertheless, our data suggest that membrane dynamics may play an important role in VLDL lipidation in vivo. Interestingly, a recent proteomic study identified Lpcat3 as a component of the VLDL transport vesicle, indicating that Lpcat3 travels with primordial VLDL particles as they bud from the ER and move to the Golgi (*Rahim et al., 2012*). Together with prior work, our data favor a model in which Lpcat3 modifies the arachidonoyl-PC composition of both membranes and lipoprotein particles during VLDL assembly, thereby generating a local membrane environment that facilitates lipid transport and bulk lipidation.

Our prior studies showed that acute shRNA-mediated knockdown of Lpcat3 expression in liver exacerbated ER in the setting of obesity and hepatic steatosis. However, genetic deletion of Lpcat3 expression in liver did not lead to overt ER stress pathway activation, suggesting that increased abundance of monounsaturated PC species (e.g., 16:0, 18:1 PC, *Figure 5*) may partially compensate to maintain ER membrane homeostasis. It remains to be determined whether stronger lipotoxic stimuli, such as the hepatosteatosis observed in *ob/ob* mice, may yet provoke increased ER stress responses in the genetic absence of Lpcat3.

This work illustrates how manipulation of membrane composition can be used as regulatory mechanism to control metabolic pathways. A model for Lpcat3 action is presented in *Figure 12*. Lpcat3 is not only a required component of lipoprotein production; it is also a regulated one. The fact that Lpcat3 expression is dynamically regulated by LXRs in liver and intestine strongly suggests that the level of Lpcat3 enzymatic activity has evolved to respond to dietary and metabolic demands. LXR is transcriptionally activated as cellular cholesterol levels rise. Thus, the LXR-Lpcat3 pathway provides a mechanism to integrate sterol metabolism and membrane PL composition. We speculate that Lpcat3 expression is induced in response to lipid loading in order to increase ER/Golgi membrane flexibility and to facilitate efficient TG secretion (thereby unloading cells of excess lipids). In support of this idea, we found that the ability of pharmacologic LXR agonist to stimulate hepatic VLDL production was impaired in the genetic absence of Lpcat3.

Finally, these studies open the door to new strategies for pharmacologic intervention in systemic lipid metabolism. Abnormal phospholipid metabolism has been associated with several metabolic diseases. For example, lower amounts of polyunsaturated PLs have been observed in liver biopsies from nonalcoholic steatohepatitis patients (*Puri et al., 2007*), and cell membranes from patients with atherosclerotic disease tend to show decreased membrane fluidity (*Chen et al., 1995*). However, it has heretofore been difficult to draw a causal link between aberrant PL composition and the pathogenesis of metabolic diseases. Our studies provide direct evidence that alterations in

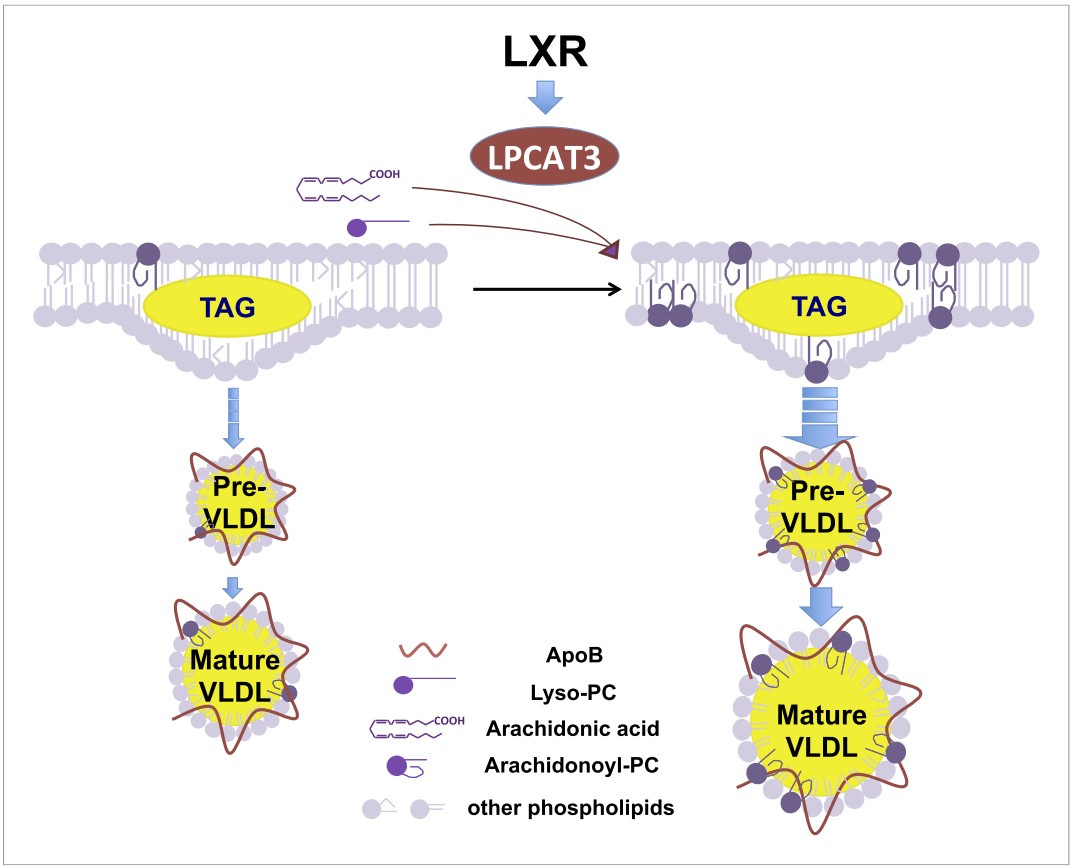

**Figure 12**. Schematic illustration of the role of the LXR-Lpcat3 pathway in VLDL lipidation. Activation of LXRs promotes the incorporation of arachidonate into intracellular membranes through the induction of Lpcat3 expression. This change in membrane composition creates a dynamic membrane environment that facilitates the transfer of TG synthesized on ER and/or in cytosolic LD to nascent apoB–containing lipoproteins particles, leading to the efficient lipidation of apoB–containing lipoproteins.

arachidonoyl PC abundance impair lipoprotein metabolism in liver and intestine in vivo. Recently, a human GWAS reported a highly significant association between *LPCAT3* and the phospholipid composition of red blood cell membranes, indicating that Lpcat3 is also a key regulator of phospholipid composition in humans (*Tintle et al., 2014*). Therefore, an improved understanding of the mechanisms by which Lpcat3 and arachidonoyl PCs regulate lipid homeostasis could lead to new approaches to metabolic diseases. We speculate that small molecule inhibitors of Lpcat3 could be of potential utility for lowering plasma lipid levels in hyperlipidemic individuals. In pursuing such a strategy, one would need to monitor the impact of Lpcat3 inhibition on liver TG stores, hepatic inflammation and lipid malabsorption, in addition to the plasma lipids. However, modest increases in liver TG stores, even if they occurred in humans, might not doom such a strategy. Inhibiting apoB synthesis with antisense compounds has found a place in the management of some cases of hyperlipidemia, despite modest increases in liver TG stores.

## Materials and methods

### Gene expression

Total RNA was isolated from cells and tissues with Trizol (Invitrogen, Carlsbad, CA). cDNA was synthesized, and gene expression was quantified by real-time PCR with SYBR Green (Diagenode, Denville, NJ) and an ABI 7900. Gene expression levels were normalized to 36B4 or GAPDH. Primer sequences are listed in *Table 4*. For microarray experiments, RNA was pooled from n = 5 biological replicates and processed in the UCLA Microarray Core Facility with Gene-Chip Mouse Gene 430.2

**Table 4**. Quantitative real-time PCR primer sequences

| Murine qPCR primers | Forward | Reverse |
| --- | --- | --- |
| CD36 | TTGAAAAGTCTCGGACATTGAG | TCAGATCCGAACACAGCGTA |
| PLPT | GTCTAAAATGAATATGGCCTTCG | CCAGAAGTGATGAACGTGGA |
| DGAT1 | TTCCGCCTCTGGGCATT | GCCCACAATCCAGGCCA |
| DGAT2 | GGCGCTACTTCCGAGACTAC | TGGTCAGCAGGTTGTGTGTC |
| APOA4 | ACCCAGCTAAGCAACAATGC | TGTCCTGGAAGAGGGTACTGA |
| APOE | GACTTGTTTCGGAAGGAGCTG | CCACTCGAGCTGATCTGTCA |
| AOPB | CTGAACATCAAGAGGGGCATC | GGTAACCTGAGTTGAGCAGTTT |
| MTTP | GGCAGTGCTTTTTCTCTGCT | TGAGAGGCCAGTTGTGTGAC |
| PDK4 | ACCGAAGAACCTGGCGAAG | TGATCCCGTAAAATGTCAGGC |
| CD68 | GACCTACATCAGAGCCCGAGT | CGCCATGAATGTCCACTG |
| CCL6 | TCTTTATCCTTGTGGCTGTCC | TGGAGGGTTATAGCGACGAT |
| ACSL3 | TCTAGGAGTGAAGGCCAACG | GCAATATCTGAGGGCAGTGG |
| ACSL5 | AACCAGTCTGTGGGGATTGAG | CGTCTTGGCGTCTGAGAAGTA |
| MOGAT2 | TCTTCCAGTACAGCTTTGGCCTCA | TGATATAGCGCTGATGAAGCCGGT |
| FABP1 | AGTACCAATTGCAGAGCCAGGAGA | GACAATGTCGCCCAATGTCATGGT |
| FABP2 | AGAGGAAGCTTGGAGCTCATGACA | TCGCTTGGCCTCAACTCCTTCATA |
| FATP1 | CGCTTTCTGCGTATCGTCTG | GATGCACGGGATCGTGTCT |
| LPCAT3 | GGCCTCTCAATTGCTTATTTCA | AGCACGACACATAGCAAGGA |
| LPCAT1 | GTGCACGAGCTGCGACT | GCTGCTCTGGCTCCTTATCA |
| LPCAT2 | TGTACTAATCGCTCCTGTTTGATT | CACTGGAACTCCTGGGATG |
| LPCAT4 | TTCGGTTTCAGAGGATACGACAA | AATGTCTGGATTGTCGGACTGAA |
| 36B4 | AGATGCAGCAGATCCGCAT | GTTCTTGCCCATCAGCACC |
| GAPDH | TGTGTCCGTCGTGGATCTGA | CCTGCTTCACCACCTTCTTGAT |

Arrays (Affymetrix, Santa Clara, CA). Data analysis was performed with GenespringGX (Agilent, Santa Clara, CA). The GEO accession numbers for the microarray data are GSE65352 and GSE65353.

## Protein analysis

Cells and tissue lysates were prepared by homogenization in RIPA buffer (50 mM Tris–HCl, pH 7.4; 150 mM NaCl, 1% NP-40, 0.5% sodium deoxycholate, 0.1% SDS) supplemented with protease and phosphatase inhibitors (Roche Molecular Biochemicals). Lysates were cleared by centrifugation. Plasma samples were diluted with RIPA buffer. Protein lysate were then mixed with NuPAGE LDS Sample Buffer, size-fractionated on 4–12% Bis-Tris Gels (Invitrogen), transferred to hybond ECL membrane (GE Healthcare, Piscataway, NJ), and incubated with an apoB antibody (Abcam, Cambridge, MA). Primary antibody binding was detected with a goat anti-rabbit secondary antibody and visualized with chemiluminescence (ECL, Amersham Pharmacia Biotech).

## Generation of Lpcat3 knockout mice

A conditional knockout allele for *Lpcat3* was generated with a sequence replacement 'knock-out first/ conditional-ready' gene-targeting vector. The vector was electroporated into JM8A1.N3 ES cell line from C57BL/6N mice. Positive clones were identified by genotyping and long-range PCR at both the 5′ and 3′ ends. Two targeted ES cell clones were injected into C57BL/6 blastocysts to generate chimeric mice. High-percentage male chimeras were obtained, and resulting chimeras bred with female C57BL/6 mice to obtain heterozygous knockout mice (*Lpcat3*[+/−]). *Lpcat3*[+/−] mice were intercrossed to produce homozygous knockout mice (*Lpcat3*[−/−]). To create a conditional knockout

allele, *Lpcat3$^{+/-}$* mice were mated with mice expressing a Flpe recombinase deleter transgene (Jackson Laboratory, Bar Harbor, Maine). That transgene excises the gene-trapping cassette in intron 2 of *Lpcat3*, producing a conditional knockout allele containing loxP sites in intron 2 and intron 3. *Lpcat3$^{Fl/Fl}$* mice were crossed with albumin-*Cre* or villin-*Cre* transgenic mice from The Jackson Laboratory (Bar Harbor, ME).

## Triglyceride secretion assay in mice

Mice were fasted for 4 hr and then injected intravenously with 500 mg/kg body weight of tyloxapol (Fisher, Pittsbugh, PA). Blood was subsequently collected, and the plasma separated by centrifugation for the measurement of TGs.

## Animal studies

*Lpcat3$^{-/-}$*, *Lpcat3$^{fl/fl}$*, *Lpcat3$^{fl/fl}$; Albumin-Cre*, and *Lpcat3$^{fl/fl}$; Villin-Cre* mice were generated as described above. All the mice were housed under pathogen-free conditions in a temperature-controlled room with a 12-hr light/dark cycle. Mice were fed a chow diet, a western diet (Research Diets #D12079B), or a high-sucrose diet (Research Diets #D07042201). Liver tissues were collected and frozen in liquid nitrogen and stored at −80°C or fixed in 10% formalin. Blood was collected by retro-orbital bleeding, and the plasma was separated by centrifugation. Small intestines were excised and cut into three segments with length ratios of 1:3:2 (corresponding to duodenum, jejunum and ileum). For adenoviral infections, VQAd-CMV Cre/eGFP and VQAd-Empty eGFP were purchased from Viraquest. 8- to 10-week-old male *Lpcat3$^{fl/fl}$* mice were injected (via the tail vein) with $2 \times 10^{9}$ plaque-forming units (pfu). Mice were sacrificed or used for TG secretion studies 3–4 weeks after the adenovirus injection. Blood was collected from mice by cardiac puncture. Plasma lipids were measured with the Wako L-Type TG M kit, the Wako Cholesterol E kit; and the Wako HR series NEFA-HR(2) kit. Tissue lipids were extracted with Bligh-Dyer lipid extraction (*Bligh and Dyer, 1959*) and measured with the same enzymatic kits. Plasma fast protein liquid chromatography (FPLC) lipoprotein profiles were performed in the Lipoprotein Analysis Laboratory of Wake Forest University School of Medicine. Tissue histology was performed in the UCLA Translational Pathology Core Laboratory. Animal experiments were conducted in accordance with the UCLA Animal Research Committee.

## Lipid analyses

Liver tissue and plasma were snap frozen at the temperature of liquid nitrogen. Liver was homogenized on ice in phosphate buffered saline. Plasma or liver homogenates were subsequently subjected to a modified Bligh-Dyer lipid extraction (*Bligh and Dyer, 1959*) in the presence of lipid class internal standards including eicosanoic acid, 1-0-heptadecanoyl-sn-glycero-3-phosphocholine, 1,2-dieicosanoyl-sn-glycero-3-phosphocholine, cholesteryl heptadecanoate, and 1,2-ditetradecanoyl-sn-glycero-3-phosphoethanolamine (*Demarco et al., 2013*). Lipid extracts were diluted in methanol/chloroform (4/1, vol/vol) and molecular species were quantified using electrospray ionization mass spectrometry on a triple quadrupole instrument (Themo Fisher Quantum Ultra) employing shotgun lipidomics methodologies (*Han and Gross, 2005*). Lysophosphatidylcholine molecular species were quantified as sodiated adducts in the positive ion mode using neutral loss scanning for 59.1 amu (collision energy = −28 eV). PC molecular species were quantified as chlorinated adducts in the negative ion mode using neutral loss scanning for 50 amu (collision energy = 24 eV). Phosphatidylethanolamine molecular species were first converted to fMOC derivatives and then quantified in the negative ion mode using neutral loss scanning for 222.2 amu (collision energy = 30 eV). Cholesteryl ester molecular species were quantified as sodiated adducts in the positive ion mode using neutral loss scanning of 368.5 amu as previously described (*Bligh and Dyer, 1959*; *Bowden et al., 2011*). Individual molecular species were quantified by comparing the ion intensities of the individual molecular species to that of the lipid class internal standard with additional corrections for type I and type II $^{13}C$ isotope effects (*Han and Gross, 2005*). Triacylglycerol fatty acid composition was determined as follows. Samples were subjected to Bligh-Dyer extraction in the presence of the internal standard, triheptadecenoin, followed by thin layer chromatographic purification of triacylglycerol using silica gel G plates with petroleum ether/ethyl ether/acetic acid; 80/20/1 as mobile phase. Purified triacylglycerol was then subjected to fatty acid methanolysis, and fatty acid methyl esters were then determined using gas chromatography with flame ionization detection as previously described (*Ford and Gross, 1988*, *1989*).

## Membrane dynamics

Membrane dynamics was analyzed as described (*Golfetto et al., 2013*). Briefly, primary hepatocytes from *Lpcat3*[fl/fl] and *Lpcat3*[fl/fl]; *Albumin-Cre* mice were isolated as described (*Rong et al., 2013*). Cells were incubated with 1.8 mM Laurdan (6-dodecanoyl-2-dimethylaminonaphthalene; Invitrogen) at 37°C for 30 min. Cells were rinsed with phosphate-buffered saline (PBS), and fresh culture medium was added. Spectral data were acquired with a Zeiss LSM710 META laser scanning microscope coupled to a 2-photon Ti:Sapphire laser (Mai Tai, Spectra Physics, Newport Beach, CA) producing 80-fs pulses at a repetition of 80 MHz with two different filters: 460/80 nm for the blue channel and 540/50 nm for the green channel. Spectral data were processed by the SimFCS software (Laboratory for Fluorescence Dynamics). The GP value was calculated for each pixel using the two Laurdan intensity images (460/80 nm and 540/50 nm). The GP value of each pixel was used to generate the pseudocolored GP image. GP distributions were obtained from the histograms of the GP images.

## Organelle isolation

The Golgi fraction was isolated as described (*Vance and Vance, 1988*). Briefly, fresh liver tissue was homogenized with a motor driven Potter-Elvehjem homogenizer in homogenization buffer (37.5 mM TRIS-maleate, pH 7.4; 0.5 M sucrose; 1% dextran; and 5 mM $MgCl_2$). After an initial centrifugation at 5000×*g* for 15 min, most of the supernatant was removed and the yellow-brown portion (approximately the upper one-third) of the pellet was suspended in homogenization buffer, layered over 2.7 ml of 1.2 M sucrose and spin for 30 min at 100,000×*g* in a swinging bucket rotor (SW50.1). The Golgi fraction was collected at the buffer/1.2 M sucrose interface. The Golgi was suspended in distilled water and pelleted by centrifugation at 5500×*g* for 20 min. Aliquots of the Golgi fraction were used to measure TG and protein concentration. Golgi apoB was analyzed as described (*Gusarova et al., 2003*; *Li et al., 2012*). Briefly, the luminal contents were released from Golgi fraction by treatment with 0.1 M sodium carbonate (pH 11) and deoxycholic acid (0.025%) for 30 min at room temperature. Bovine serum albumin (BSA) was added to a final concentration of 5 mg/ml, followed by centrifugation (50,000 rpm in a Beckman SW60 rotor for 1 hr at 4°C) to remove membranes. The luminal contents (supernatant) were adjusted to pH 7.4 with acetic acid, adjusted to a sucrose concentration of 12.5% (wt/vol), and placed on the top of a step gradient consisting of 1.9 ml of 49% sucrose and 1.9 ml of 20% sucrose. Next, 2.8 ml of PBS was layered on the top of the supernatants. All solutions contained protease inhibitors. After centrifugation at 35,000 rpm for 65 hr at 10°C in a Beckman SW41 rotor, 12 fractions with densities ranging from 1.0 to 1.125 g/ml were collected from the top of the tube. ApoB protein from each fraction was analyzed by western blotting.

## Electron microscopy

For VLDL isolation, 400 μl plasma from fasted mice was put into the bottom of a 1-ml thick-walled polycarbonate tube, overlaid with 600 μl of 1.006 g/ml KBr solution, and centrifuged at 100,000 rpm at 16°C for 2 hr in a TLA 120.2 rotor. 300 μl from the top of the tube was collected as the VLDL fraction. For EM, 5 μl of the VLDL preparation was applied to carbon-coated copper grids and stained with 2.0% uranyl acetate for 15 min. Grids were visualized with a JEOL 100CX transmission electron microscope.

For EM of liver samples, animals were perfused through the left ventricle of the heart with a fixative of 1.5% glutaraldehyde, 4% polyvinylpyrrolidone, and 0.05% calcium chloride in 0.1 M sodium cacodylate buffer pH 7.4, after an initial flush with 0.1 M sodium cacodylate buffer, pH 7.4. The imidazole-buffered osmium tetroxide procedure described by Angermuller and Fahimi (*Angermuller and Fahimi, 1982*) was used to stain for lipids. Tissue was en block stained in aqueous uranyl acetate, dehydrated, infiltrated and embedded in LX-112 resin (Ladd Research Industries, Burlington, VT). Samples were ultrathin sectioned on the Reichert Ultracut S ultramicrotome and counter stained with 0.8% lead citrate. Grids were examined on a JEOL JEM-1230 electron microscope (JEOL USA, Inc., Peabody, MA) and photographed using the Gatan Ultrascan 1000 digital camera (Gatan Inc., Warrendale, PA).

## Acknowledgements

We thank Jinkuk Choi and Ito Ayaka for technical support. This work was supported by grants HL090553, DK063491, HL030568, HL074214, GM103540, GM076516 and AHA Fellowship 13PRE17150049. PT is an Investigator of the Howard Hughes Medical Institute.

## Additional information

### Competing interests

SGY: Reviewing editor, *eLife*. PT: Reviewing editor, *eLife*. The other authors declare that no competing interests exist.

### Funding

| Funder | Grant reference | Author |
|---|---|---|
| Howard Hughes Medical Institute (HHMI) | Investigator | Peter Tontonoz |
| National Institutes of Health (NIH) | HL090553 | Stephen G Young, Peter Tontonoz |
| National Institutes of Health (NIH) | DK063491 | Peter Tontonoz |
| National Institutes of Health (NIH) | HL030568 | Peter Tontonoz |
| National Institute of General Medical Sciences (NIGMS) | GM103540 | Enrico Gratton |
| National Institute of General Medical Sciences (NIGMS) | GM076516 | Enrico Gratton |
| National Institutes of Health (NIH) | HL074214 | David A Ford |
| American Heart Association (AHA) | 13PRE17150049 | Xin Rong |

The funders had no role in study design, data collection and interpretation, or the decision to submit the work for publication.

### Author contributions

XR, BW, Conception and design, Acquisition of data, Analysis and interpretation of data, Drafting or revising the article; MMD, PNH, JSW, Acquisition of data, Analysis and interpretation of data; EG, Conception and design, Analysis and interpretation of data; SGY, PT, Conception and design, Analysis and interpretation of data, Drafting or revising the article; DAF, Acquisition of data, Analysis and interpretation of data, Drafting or revising the article

### Author ORCIDs

Peter Tontonoz, http://orcid.org/0000-0003-1259-0477

### Ethics

Animal experimentation: This study was performed in strict accordance with the recommendations in the Guide for the Care and Use of Laboratory Animals of the National Institutes of Health. All of the animals were handled according to approved institutional animal care and use committee (IACUC) protocols (99-131 and 2003-166) of the University of California Los Angeles.

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
