## [Decision Letter]

Thank you for sending your work entitled “Lpcat3-dependent production of arachidonyl phospholipids is a key determinant of triglyceride secretion” for consideration at *eLife*. Your article has been favorably evaluated by Vivek Malhotra (Senior editor) and three reviewers, one of whom, Tobias Walther, has served as a guest Reviewing editor.

The Reviewing editor and the other reviewers discussed their comments before we reached this decision, and the Reviewing editor has assembled the following comments to help you prepare a revised submission.

All reviewers noted the high quality of data, as well as the interesting findings presented. For a number of experiments, the reviewers feel there is a need for clarification.

First, the interpretation of Lpcat3 deficiency leading to membrane fluidity changes which in turn leads to the observed lipoprotein secretion defects is interesting, but not convincingly and fully shown here. Laurdan staining could reflect other changes in the membrane besides fluidity (curvature, order…) and it is unclear really how this would lead to the phenotype described. The authors should consider revising this point and elaborate the Discussion. For example, in the Discussion the authors state that lipid changes “increase membrane fluidity and curvature”(does this mean the ER structure changes? Where does the curvature change? How do they know that happens?).

Second, the experiment fractionating Golgi and microsomes should be quantified and clarified. Was tauro-cholate or another detergent used here? Earlier data from the Olofsson laboratory indicate that in the absence of the detergent significant amounts of apoB lipoproteins associated with inner leaflets of microsomes. This might indicate that the authors in their experiments may select for a subpopulation of particles. It would also be of interest to see whether the ER and Golgi apoB-lipoproteins both were skewed in buoyant density to denser particles in the absence of Lpcat3. While the reviewers agree that most lipid is added to VLDL in the Golgi, it is also possible that there are some larger particles also formed in the ER (as implied by classic EM studies by Bob Hamilton) and they may be similarly affected by Lpcat3 deficiency.Third, since the group has previously suggested that ablation of hepatic Lpcat3 expression promotes inflammatory response and ER stress it is important to address whether these conditions might cause of the phenotype observed in liver-specific Lpcat3 knockout mice.

---

## [Author Response]

*First, the interpretation of Lpcat3 deficiency leading to membrane fluidity changes which in turn leads to the observed lipoprotein secretion defects is interesting, but not convincingly and fully shown here. Laurdan staining could reflect other changes in the membrane besides fluidity (curvature, order…) and it is unclear really how this would lead to the phenotype described. The authors should consider revising this point and elaborate the Discussion. For example, in the Discussion the authors state that lipid changes “increase membrane fluidity and curvature”(does this mean the ER structure changes? Where does the curvature change? How do they know that happens?)*.

We acknowledge the limitations of the Laurdan studies and we have revised our Discussion to be more circumspect in the conclusions drawn from this data.

*Second, the experiment fractionating Golgi and microsomes should be quantified and clarified. Was tauro-cholate or another detergent used here? Earlier data from the Olofsson laboratory indicate that in the absence of the detergent significant amounts of apoB lipoproteins associated with inner leaflets of microsomes. This might indicate that the authors in their experiments may select for a subpopulation of particles. It would also be of interest to see whether the ER and Golgi apoB-lipoproteins both were skewed in buoyant density to denser particles in the absence of Lpcat3. While the reviewers agree that most lipid is added to VLDL in the Golgi, it is also possible that there are some larger particles also formed in the ER (as implied by classic EM studies by Bob Hamilton) and they may be similarly affected by Lpcat3 deficiency*.

We thank the reviewers for this insightful comment. We have responded in two ways. First, we used detergent (0.025% deoxycholic acid) when we isolated the Golgi luminal contents. We have now included these missing details in the Methods. Second, we have added new EM data that directly assess the size of nascent lipoprotein particle in the ER and Golgi. These studies provide definitive support for our conclusion that the particles are less lapidated in the absence of Lpcat3.

*Third, since the group has previously suggested that ablation of hepatic Lpcat3 expression promotes inflammatory response and ER stress it is important to address whether these conditions might cause of the phenotype observed in liver-specific Lpcat3 knockout mice*.

This was also an excellent point. We do not see prominent induction of ER stress responses in the Lpcat3 KO mice, and we believe that this is related to the difference between the experimental approaches (acute Lpcat3 knockdown vs genetic deletion). We have now added this data to the revised manuscript. We point out that we do in fact observe increased inflammation in the Lpcat3 KO mice in the setting of western diet feeding, and these data are included in the paper. The fact that the lipoprotein production defect is observed in the absence of inflammation (e.g. on chow diet), supports the conclusion that this defect is not secondary to inflammatory changes.